# Learning from Comparison: Constrained Projection Policy Optimization for Pareto-Front Improvement

**Jintao Li** [1]  **Maowen Tang** [1]  **Yongji Long** [1]  **Weixuan Liu** [1]  **Yanlang Zheng** [1]  **Sicheng He** [1]  **Ao-Jin Li** [1]  **Shui Yu** [1 *]
**Yun Li** [1 *]

## Abstract

Constrained multi-objective reinforcement learning aims to discover a diverse set of feasible trade-offs, yet scalarization and signed, normalized group-relative advantages can be brittle under objective-scale drift, near-ties, and feasibility scarcity. We propose constrained projection policy optimization (CoPro), which alternates between an E-step moment projection and an M-step policy projection. In the E-step, we solve a Kullback-Leibler (KL)-regularized, moment-constrained projection over each sampled group to compute a nonnegative reweighting distribution ($q^*$) that promotes feasible Pareto-front (PF) progress, preserves feasibility anchors, and suppresses ambiguous near-ties. This E-step admits a closed-form exponential-family solution and guarantees strictly positive probability mass on feasible anchors whenever feasible candidates appear in the group. In the M-step, we project the policy toward $q^*$ via weighted maximum likelihood with a trust-region regularizer, yielding a PF-aligned update direction from comparisons without handcrafted reward shaping. Empirically, CoPro improves feasible PF quality and robustness on constrained multi-objective benchmarks for large language model tool use and analog circuit design tasks; code is available at `CoPro`.

## 1. Introduction

Modern optimization and design problems are rarely governed by a single metric. Across domains such as robotics (Khanda et al., 2025), chip design (Li et al., 2025a;c;b; 2026a;b), and large language model training (Liu et al.,

---
[*]Corresponding authors. [1]Shenzhen Institute for Advanced Study, UESTC, Shenzhen, China. Correspondence to: Shui Yu <yushui@uestc.edu.cn>, Yun Li <yun.li@ieee.org>.

*Proceedings of the 43rd International Conference on Machine Learning*, Seoul, South Korea. PMLR 306, 2026. Copyright 2026 by the author(s).

2026), policies must satisfy hard constraints while balancing multiple competing objectives. The practical goal is therefore to rapidly discover a diverse set of feasible trade-offs, namely a high-quality feasible Pareto front (PF), under limited evaluation budgets and substantial observation noise (Huang et al., 2022).

A common practice is to encode the relative importance of objectives via a preference vector and learn a preference-conditioned mapping by maximizing a scalarized reward (Yang et al., 2025; Wirth et al., 2017). While simple, scalarization can be brittle in constrained multi-objective regimes (Achiam et al., 2017). Objective scales may drift across tasks or contexts, constraints often make the feasible region highly non-uniform, and Pareto-relevant progress is frequently small and context-dependent (e.g., an update matters primarily because it expands an under-covered region of the feasible PF). As a result, learning signals become sensitive to weights and penalty schedules and may over-focus on particular trade-offs rather than improving PF coverage and quality (Yang et al., 2025; Tessler et al., 2018).

Motivated by within-group relative learning (Shao et al., 2024), we take a comparative perspective. When a policy generates a batch of candidates within the same context or across short rollouts, their outcomes naturally induce rich *relative* relations: which candidates are feasible, which are dominated, and which improve coverage of the current PF the most. Such comparison signals are abundant and less tied to absolute objective scales, enabling advantage construction from within-group feedback. However, naively mapping comparisons to *signed* scalar advantages via heuristic normalization is unstable under constraints and multiple objectives (Liu et al., 2026).

Two failure modes are particularly pronounced in constrained multi-objective comparisons: (i) **Sign-flip cancellation under near-ties.** Pareto partial order yields many non-dominated and near-indifferent candidates, especially under noise or early search (Li et al., 2014). Heuristic signed advantages can assign opposite signs to highly similar samples, creating destructive interference on shared structures and producing weak or oscillatory updates. (ii) **Loss of feasible anchors under feasibility scarcity.** Feasible

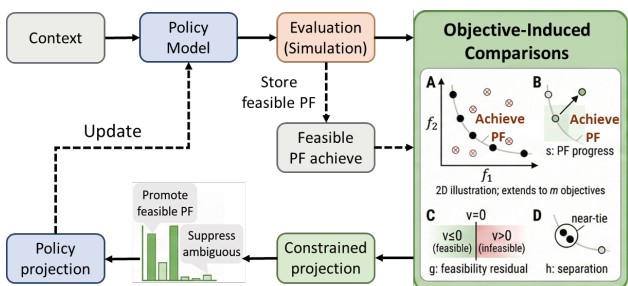

*Figure 1.* CoPro extracts within-group relative signals and computes a moment-constrained target distribution $q^*$, which guides the policy update via weighted projection.

samples may be rare and unevenly distributed (Ding et al., 2023). Without explicit anchoring, within-group reweighting can be dominated by infeasible candidates, making progress highly sensitive to penalty tuning and causing collapse toward feasibility-chasing or noisy dense surrogates. These issues reflect a deeper mismatch: multi-objective and constraint-coupled comparisons carry structured information, but signed scalar weights compress it into a global verdict that is unstable under near-ties and scarce feasibility.

To address this mismatch, we propose Constrained Projection Policy Optimization (CoPro), shown in Fig. 1. CoPro replaces heuristic signed advantages with a nonnegative reweighting distribution $q^*$ computed by a KL-regularized, moment-constrained projection over each group. The projection uses bounded group-level statistics capturing (i) feasible PF progress, (ii) feasibility residuals, and (iii) within-group distinctiveness to de-emphasize near-ties. We then perform KL-regularized projection to obtain an optimal weighting distribution $q^*$, which admits a closed-form exponential-family solution. Finally, we update the policy by projecting it toward $q^*$ via weighted maximum likelihood, yielding a direct, PF-aligned update direction derived from within-group comparisons.

1. **Moment-constrained distribution projection for comparative policy optimization.** We introduce CoPro, which replaces signed scalar advantages by a KL-regularized moment projection that produces a nonnegative target distribution $q^*$, enabling comparative learning under hard constraints and multiple objectives.

2. **Closed-form E-step and sample-level anchoring.** We derive the exponential-family form of $q^*$ and prove a sample-level feasibility-anchor guarantee: whenever a group contains feasible candidates, the feasibility moment constraint enforces strictly positive mass on feasible anchors. We also show how nonnegative reweighting reduces sign-flip-induced cancellation, while a distinctiveness moment prevents over-concentration on near-ties.

3. **Evaluation on constrained multi-objective tasks.** We evaluate CoPro on constrained multi-objective benchmarks spanning LLM reasoning optimization and analog integrated circuit (IC) design. Across both settings, CoPro improves feasible-PF quality over competitive baselines and exhibits stronger robustness to objective-scale variation.

**Conflict of Interest Disclosure.** The authors declare no financial conflicts of interest related to this work.

## 2. Preliminaries

### 2.1. Related Work

Constrained multi-objective reinforcement learning (MORL) seeks a set of Pareto-optimal behaviors while satisfying safety or feasibility constraints. A dominant line of MORL represents trade-offs with a preference vector and learns a preference-conditioned policy (Yang et al., 2025), typically linear scalarization via weighted inner products, implemented through specialized Bellman operators or actor-critic updates (Basaklar et al., 2022; Lu et al., 2023; Cai et al., 2023; Kyriakis et al., 2022). Beyond linear scalarization, nonlinear schemes such as Chebyshev scalarization and population-based evolutionary strategies have been used to cover more complex trade-off sets (Xu et al., 2020; Shianifar et al., 2025; Kim et al., 2025).

On the constraint-handling side, existing methods commonly adopt either primal-dual updates with multiplier adaptation (Gabbianelli et al., 2024; Chen et al., 2024; Bai et al., 2022) or primal trust-region-style constrained policy updates (Xu et al., 2021; Kim et al., 2023). However, scalarization-based constrained MORL often yields brittle learning signals under scale drift and sparse Pareto improvements, motivating approaches that learn from relative, archive-referenced progress signals instead (Stooke et al., 2020; Gao et al., 2024). This motivates our departure from reward shaping toward learning feasible-front progress directly from objective-induced comparisons. Finally, alternatives to reward-space preferences have been explored by injecting preferences into the action distribution and extending this idea to constrained multi-objective settings (Cai et al., 2023; Huang et al., 2022); our approach is complementary in that it focuses on constructing and stabilizing PF-progress, aligned comparison signals under limited evaluation budgets.

### 2.2. Group-Relative Policy Optimization

GRPO replaces value-function-based advantage estimation with a group-relative estimator. For a conditioning context $c$, GRPO samples a group of $G$ candidates from the old policy

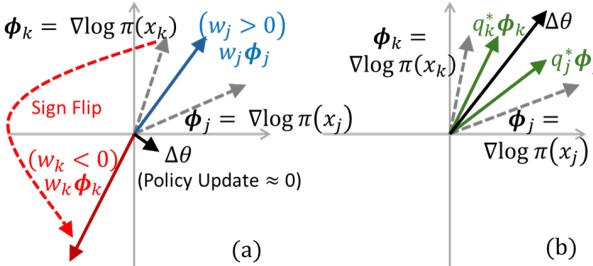

*Figure 2.* Signed scalar reweighting can induce sign-flip cancellation on shared patterns, while nonnegative reweighting mitigates sign-flip-induced interference.

$\pi_{\theta_{\text{old}}}$:

$$x_i \sim \pi_{\theta_{\text{old}}}(\cdot \mid c), \quad i = 1, \ldots, G. \tag{1}$$

A reward model assigns a scalar reward $R(x_i)$, and GRPO computes the standardized within-group advantage

$$A_i = \frac{R(x_i) - \text{mean}(\{R(x_j)\}_{j=1}^G)}{\text{std}(\{R(x_j)\}_{j=1}^G)}. \tag{2}$$

GRPO then optimizes a PPO-style clipped surrogate with a KL penalty. For gradient-direction analysis, we consider the simplified objective where clipping is inactive, and the KL penalty is omitted:

$$\tilde{\mathcal{J}}_{\text{GRPO}}(\theta) = \mathbb{E}_{c,\, x_i \sim \pi_{\theta_{\text{old}}}} \left[ \frac{1}{G} \sum_{i=1}^G r_i(\theta)\, A_i \right], \tag{3}$$

where $r_i(\theta)$ is the likelihood ratio between $\pi_\theta$ and $\pi_{\theta_{\text{old}}}$. The corresponding policy gradient is

$$\nabla_\theta \tilde{\mathcal{J}}_{\text{GRPO}}(\theta) = \frac{1}{G} \sum_{i=1}^G A_i\, r_i(\theta)\, \nabla_\theta \log \pi_\theta(x_i \mid c). \tag{4}$$

In GRPO-style updates, $\pi_\theta$ is initialized from $\pi_{\theta_{\text{old}}}$ and typically constrained to remain within a small KL neighborhood of $\pi_{\theta_{\text{old}}}$. Within such a trust region, $r_i(\theta)$ admits a first-order expansion around $\theta_{\text{old}}$, implying $r_i(\theta) \approx 1$ up to higher-order terms in the step size. Therefore, under the common early-update approximation $r_i(\theta) \approx 1$, we analyze directional conflicts by isolating the effect of group-level weights. However, straightforward application of Eq. (4) in multi-objective settings creates conflicting gradient updates when objectives disagree, as the scalar Advantage $A_i$ collapses the vector structure of the rewards.

## 3. Signed Scalar Reweighting Induces Gradient Cancellation

**Signed compression yields untrusted local updates.** Group-relative learning relies on within-group comparisons.

In multi-objective constrained settings, the comparison signal is intrinsically structured: objectives induce a partial order, while constraints create a feasible-infeasible boundary with severe feasibility scarcity. A failure mode arises when this structured signal is compressed into a single signed scalar weight per sample.

Let $x_i \sim \pi_{\theta_{\text{old}}}(\cdot \mid c)$ be a group under context $c$, and define the score-function gradient

$$\phi_i := \nabla_\theta \log \pi_\theta(x_i \mid c). \tag{5}$$

GRPO assigns a signed scalar weight $w_i$ and updates by

$$\nabla_\theta \tilde{\mathcal{J}}_{\text{GRPO}} \approx \frac{1}{G} \sum_{i=1}^G w_i\, \phi_i, \tag{6}$$

using the standard approximation $r_i(\theta) \approx 1$. In multi-objective constrained comparisons, the sign of $w_i$ is a global verdict on the entire candidate $x_i$ after aggregating heterogeneous factors, and is not attributable to any specific local pattern $\omega$ inside $x_i$.

For intuition, write the score gradient as a sum over local patterns:

$$\phi_i = \sum_{\omega \in x_i} \phi_i(\omega). \tag{7}$$

This decomposition is a conceptual device for shared substructures in generated responses or circuit designs; Appendix E gives the corresponding formal shared-component model. The induced update on a shared pattern $\omega$ is then

$$\Delta(\omega) \propto \sum_{i:\omega \in x_i} w_i\, \phi_i(\omega). \tag{8}$$

Since $w_i$ can flip due to components of $x_i$ unrelated to $\omega$, the sign acts as a high-variance gate on updates to $\omega$, amplifying cancellation on reusable structures (Yu et al., 2020).

Two properties make this effect systematic: (i) Pareto non-dominance yields many near-ties, so the sign of an aggregated scalar weight becomes sensitive to noise and tie-breaking; (ii) feasibility scarcity makes the aggregate weight dominated by feasibility-related surrogates or dense fallbacks, so the sign reflects feasibility proximity rather than objective improvement. Both mechanisms increase sign flips across candidates that share useful patterns, producing destructive interference in (8), as shown in Fig. 2.

**Cancellation index.** We quantify cancellation by comparing the norm of the signed aggregate update to the sum of per-sample magnitudes:

$$\mathcal{C}(w) = 1 - \frac{\left\| \sum_{i=1}^G w_i \phi_i \right\|}{\sum_{i=1}^G |w_i|\, \|\phi_i\| + \varepsilon}, \quad \mathcal{C}(w) \in [0, 1], \tag{9}$$

where $\varepsilon > 0$ is a small constant for numerical stability. Large $\mathcal{C}(w)$ indicates strong cancellation in (6) despite large per-sample gradient magnitudes. We report $\mathcal{C}(w)$ in Sec. 5 to diagnose cancellation and its reduction under nonnegative reweighting empirically.

The analysis isolates a structural mismatch: a signed scalar weight couples multi-objective and constraint information into a global sign that is not locally attributable, inducing sign-flip cancellation on shared patterns. This motivates replacing signed weights by a nonnegative reweighting distribution $q^* \in \Delta^G$ obtained by the KL-regularized moment projection in Sec. 4.2.

# 4. Constrained Group-Relative Policy Optimization via Distribution Projection

We retain GRPO's group sampling and formalize CoPro as a per-group constrained distribution projection (E-step) followed by a weighted policy projection (M-step), as shown in Fig. 3.

## 4.1. From Signed Advantages to Constrained Reweighting

**Comparative multi-objective constrained feedback.** In our setting, evaluating a candidate $x$ returns a multi-objective vector and a nonnegative constraint violation:

$$y(x) = \big(f(x) \in \mathbb{R}^m, \ v(x) \geq 0\big), \qquad (10)$$

where $f$ denotes the objective vector and $v(x) = 0$ indicates feasibility. Let $\mathcal{A}_{\mathcal{PF}}$ denote the current feasible-PF archive. A group $\{x_i\}$ naturally induces rich *relative* relations without additional supervision. We summarize these relations by (i) a PF-progress-discriminative score $S(x \mid \mathcal{A}_{\mathcal{PF}})$ and (ii) a signed feasibility residual $g(x)$ used by moment constraints in Sec. 4.2. We only require that the per-sample statistics introduced below are *bounded* scalars; a concrete instantiation consistent with our implementation (including the construction of $S$, the mapping from $v$ to $g$, and the distinctiveness mapping) is provided in Appendix A.

**PF-progress score.** A concrete instantiation used throughout this work combines (i) a sparse front-expansion signal with (ii) a dense fallback signal to avoid vanishing updates when PF progress is rare:

$$S(x \mid \mathcal{A}_{\mathcal{PF}}) = \lambda_{\mathrm{pf}}\, p(x \mid \mathcal{A}_{\mathcal{PF}}) + \lambda_{\mathrm{dense}}\, d(x \mid \mathcal{A}_{\mathcal{PF}}), \ (11)$$

where $p$ measures *feasible* PF expansion relative to $\mathcal{A}_{\mathcal{PF}}$, and $d$ is a dense surrogate that provides within-group learning signal when $p$ is zero for most samples. Importantly, feasibility is *not* folded into $S$; it is enforced explicitly by constraints in Sec. 4.2. The exact definitions of $p$ and $d$ used in our implementation are given in Appendix A.

**Group construction.** Given a context $c$, we sample a group $\{x_i\}_{i=1}^G$ from the old policy $\pi_{\theta_{\mathrm{old}}}(\cdot \mid c)$ and evaluate each $x_i$ to obtain $y(x_i)$. We then compute three *bounded* scalars:

- **Comparative PF-progress score:** $s_i := S(x_i \mid \mathcal{A}_{\mathcal{PF}})$, optionally normalized/clipped to a bounded range for numerical stability;

- **Feasibility residual:** $g_i := g(x_i)$, where $g_i \leq 0$ for feasible anchors and $g_i > 0$ otherwise. We use a bounded mapping from $v(x_i)$ to $g_i$ in our implementation; see Appendix A.

- **Distinctiveness:** $h_i := h(x_i) \in [0, h_{\max}]$, which is *small* for near-indifferent samples and increases with comparative separation. We use a median-based mapping from within-group scores to $h_i$; see Appendix A.

A naive GRPO adaptation would replace $R(x_i)$ by $s_i$ in (2). Instead, we compute a *nonnegative* reweighting distribution $q(\cdot \mid c) \in \Delta^G$ that enforces feasibility via moment constraints in Sec. 4.2.

## 4.2. E-Step: KL-Regularized Moment Projection

Let $u \in \Delta^G$ denote a base distribution over the group. Unless stated otherwise, we use the uniform base distribution $u_i = 1/G$ over the current group or augmented group. We compute the reweighting distribution by the following KL-regularized projection:

$$q^* \in \arg\max_{q \in \Delta^G} \sum_{i=1}^G q_i s_i \ - \ \tau_{\mathrm{KL}} \, \mathrm{KL}(q \| u),$$
$$\text{s.t.} \ \sum_{i=1}^G q_i g_i \leq 0, \qquad \sum_{i=1}^G q_i h_i \geq \tau_c. \qquad (12)$$

The KL regularizer enforces a conservative improvement step; the feasibility moment constraint maintains anchors under scarce feasibility; and the distinctiveness moment constraint de-emphasizes near-ties that introduce high variance.

To avoid infeasibility when the group consists of near-ties, we set the threshold adaptively as

$$\tau_c := \rho \cdot \max_{i \in \{1,\dots,G\}} h_i, \qquad \rho \in (0,1). \qquad (13)$$

This guarantees feasibility of $\sum_i q_i h_i \geq \tau_c$ whenever $\max_i h_i > 0$. When $\max_i h_i = 0$, we set $\tau_c = 0$ so that the distinctiveness constraint is vacuous.

If $g_i > 0$ for all samples in the group, the constraint $\sum_i q_i g_i \leq 0$ is infeasible. In this case we *inject a feasibility anchor* from the archive by augmenting the group with one previously evaluated feasible design $x_{\mathrm{anc}} \in \mathcal{A}_{\mathcal{PF}}$ (hence

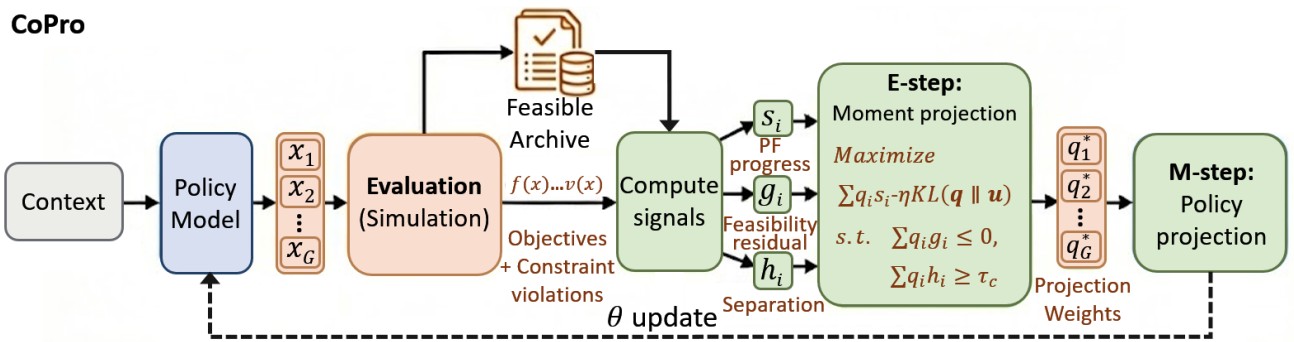

*Figure 3.* CoPro overview. For each context $c$, sample a group under policy, evaluate objectives/constraints, solve a KL-regularized moment projection to get nonnegative weights $q^*$, then update the policy by projection.

no additional environment queries), setting $(s_{\mathrm{anc}}, g_{\mathrm{anc}}, h_{\mathrm{anc}})$ accordingly, and solving (12) on the augmented group. If the archive is empty early in training, we alternatively use a relaxed feasibility moment $\sum_i q_i g_i \leq \delta_t$ with $\delta_t \downarrow 0$; see Appendix B.3 for a formal statement.

**Theorem 4.1** (Optimal reweighting structure). *Assume the constraint set in (12) is feasible for the current group and $\tau_{\mathrm{KL}} > 0$. Then (12) admits a* unique *optimum $q^* \in \Delta^G$ of the exponential-family form:*

$$
\begin{aligned}
q_i^* &= \frac{u_i \exp(a_i)}{\sum_{j=1}^G u_j \exp(a_j)}, \\
a_i &= \frac{s_i - \kappa g_i + \mu h_i}{\tau_{\mathrm{KL}}}.
\end{aligned}
\tag{14}
$$

*where $\kappa \geq 0$ and $\mu \geq 0$ are optimal Lagrange multipliers associated with the feasibility and distinctiveness moment constraints.*

The objective in (12) is strictly concave in $q$ for $\tau_{\mathrm{KL}} > 0$, and the constraints are convex. KKT conditions yield the stated exponential-family solution. A full derivation together with an efficient dual solver for $(\kappa, \mu)$ is provided in Appendix B.2.

### 4.3. Feasibility Anchors

Let $\mathcal{F} = \{i : g_i \leq 0\}$ be the feasible (or near-feasible) set and $\mathcal{I} = \{1, \ldots, G\} \setminus \mathcal{F}$ the infeasible set. Assume $\mathcal{F} \neq \emptyset$ and $\mathcal{I} \neq \emptyset$, and define $g_{\min}^- := \min_{i \in \mathcal{F}} g_i < 0$ and $g_{\min}^+ := \min_{i \in \mathcal{I}} g_i > 0$.

**Lemma 4.2** (Positive anchor mass under feasibility constraint). *For any $q \in \Delta^G$ satisfying $\sum_{i=1}^G q_i g_i \leq 0$, the total mass on $\mathcal{F}$ is lower bounded by*

$$
q(\mathcal{F}) := \sum_{i \in \mathcal{F}} q_i \geq \frac{g_{\min}^+}{g_{\min}^+ - g_{\min}^-} > 0.
\tag{15}
$$

The bound follows by minimizing $q(\mathcal{F})$ under the moment constraint and reducing to a two-point extreme distribution;

see Appendix B.1. Lemma 4.2 is a *sample-level* guarantee: whenever a group contains at least one feasible sample, the feasibility moment constraint forces strictly positive probability mass on feasible anchors. In practice, anchor injection ensures this property holds even when feasibility is initially scarce.

### 4.4. M-Step: Policy Update as KL Projection

With $q^*(\cdot \mid c)$ computed, we update the policy by projecting $\pi_\theta(\cdot \mid c)$ toward the improved sample distribution. The natural objective is the conditional KL projection with a trust-region regularizer:

$$
\begin{aligned}
\min_\theta \; \mathbb{E}_c \Big[ &\mathrm{KL}\big(q^*(\cdot \mid c) \,\|\, \pi_\theta(\cdot \mid c)\big) \\
&+ \beta \, \mathrm{KL}\big(\pi_\theta(\cdot \mid c) \,\|\, \pi_{\mathrm{ref}}(\cdot \mid c)\big) \Big],
\end{aligned}
\tag{16}
$$

where we set $\pi_{\mathrm{ref}} = \pi_{\theta_{\mathrm{old}}}$. On the finite group support, minimizing (16) is equivalent to weighted maximum likelihood:

$$
\begin{aligned}
\mathcal{L}_{\mathrm{proj}}(\theta) = \mathbb{E}_c \Big[ &-\sum_{i=1}^G q_i^* \log \pi_\theta(x_i \mid c) \Big] \\
&+ \beta \, \mathbb{E}_c \Big[ \mathrm{KL}\big(\pi_\theta(\cdot \mid c) \,\|\, \pi_{\mathrm{ref}}(\cdot \mid c)\big) \Big].
\end{aligned}
\tag{17}
$$

This EM-style update separates (i) *distribution improvement in sample space* via $q^*$ from (ii) *function approximation/projection* via $\pi_\theta$.

Let $\ell_c(\theta) := -\sum_{i=1}^G q_i^* \log \pi_\theta(x_i \mid c)$ denote the first term in (17) for group $c$. Differentiating this term yields the group-relative gradient

$$
\nabla_\theta \ell_c(\theta) = -\sum_{i=1}^G q_i^* \nabla_\theta \log \pi_\theta(x_i \mid c).
\tag{18}
$$

Replacing signed advantages by $q^* \geq 0$ removes a major source of *sign-induced cancellation* in group-relative updates. For a shared pattern $\omega$ appearing in multiple samples,

the net update on parameters governing $\omega$ takes the form

$$\nabla_\theta \mathcal{J}_{\text{shared}}(\omega) \propto \Big( \sum_{i:\omega \in x_i} \alpha_i q_i^* \Big) \nabla_\theta \log \pi_\theta(\omega), \qquad (19)$$

where $\alpha_i \geq 0$ captures the contribution of the sample $x_i$ to the gradient of $\omega$. Thus, updates on shared structure are not formed by subtracting large positive and negative terms. Moreover, the distinctiveness constraint $\sum_i q_i h_i \geq \tau_c$ de-emphasizes near-ties, reducing high-variance updates driven by ambiguous comparisons.

## 5. Experiments

We evaluate CoPro on *constrained multi-objective* problems. We first evaluate in analog IC design and then validate generality on tool-calling benchmarks.

### 5.1. Tasks

**Constrained multi-objective analog IC design.** We evaluate our method on AnalogGym (Li et al., 2025a), a real-world, open-source analog IC test suite, and validate it on **eight** distinct IC design tasks. In AnalogGym, circuit evaluations are performed with the open-source simulator Ngspice (Ngspice Development Team, 2025) using the open-source SkyWater 130 nm PDK (Google & Foundry, 2020). We optimize three objectives: $\text{FOM}_S$, $\text{FOM}_L$, and AREA, where GBW is the gain-bandwidth product, SR is the slew rate, and $C_{\text{Load}}$ is the load capacitance:

$$\text{FOM}_S = \frac{\text{GBW} \cdot C_{\text{Load}}}{\text{Power}}, \ \text{FOM}_L = \frac{\text{SR} \cdot C_{\text{Load}}}{\text{Power}}. \quad (20)$$

All remaining specifications, including stability-related requirements common in compensated amplifier design (Zhi et al., 2025), are treated as constraints. Implementation details are provided in Appendix G.

We adopt hypervolume (HV) (Zitzler & Thiele, 2002) to measure the coverage and convergence of the discovered feasible PF, a standard metric in MORL. Given a reference point $r \in \mathbb{R}^m$, define the dominated region

$$\mathcal{Q}(\mathcal{P}_T, r) := \{z \in \mathbb{R}^m \mid \exists p \in \mathcal{P}_T, \ r \preceq z \preceq p\}, \quad (21)$$

where $\preceq$ denotes element-wise inequality. The HV is defined as

$$\text{HV}(\mathcal{P}_T, r) := \int_{\mathbb{R}^m} \mathbf{1}_{\mathcal{Q}(\mathcal{P}_T, r)}(z)\, dz, \qquad (22)$$

where $\mathbf{1}_\mathcal{Q}(\cdot)$ is the indicator function.

We also report the constraint violation rate (VR) to assess the model's ability to generate feasible designs:

$$\text{VR} := \frac{1}{T} \sum_{t=1}^T \mathbf{1}[v(x_t) > 0], \qquad (23)$$

and quantify within-group gradient cancellation induced by signed weights using the gradient cancellation index (GCI) defined in Eq. (9).

**Tool calling.** We extend our method to tool-use tasks, following ToolRL (Qian et al., 2025). Models are trained to invoke external tools during generation using the same training data as ToolRL, with outputs required to follow the format in Appendix I. Training optimizes a binary format reward $R_{\text{format}} \in \{0, 1\}$ and a correctness reward $R_{\text{correct}} \in [-3, 3]$ based on tool name, argument name, and argument value matching; full definitions are provided in Appendix J. To control for gains driven by verbosity, we additionally use a length reward $R_{\text{length}} \in [0, 1]$ with a target output length $l$. Two length settings are evaluated: $l=512$ (long) and $l=200$ (short). Since our method does not rely on reward shaping, we enforce the short setting by directly constraining the output length to $l=200$ instead of using $R_{\text{length}}$. Training is implemented with VERL (Sheng et al., 2024) on Qwen-2.5-Instruct (1.5B and 3B) (Qwen et al., 2025) for 100 steps; details and hyperparameters are provided in Appendix K. Evaluation is conducted on BFCL-v3 (Patil et al., 2025), reporting mean accuracy and mean format compliance over 10 runs. We also report HV computed over the reward vector (format, correctness).

### 5.2. Main Results on AnalogGym

We compare our framework against representative baselines spanning distinct paradigms: rGNN-RL (Li & Carusone, 2023), a GNN-enhanced actor-critic RL framework; Ckt-GNN (Dong et al., 2023), a GNN-guided Bayesian-optimization (BO) approach; MA-Opt (Choi et al., 2024) and MARL (Nguyen et al., 2025), multi-agent RL optimizers; RoSE-Opt (Cao et al., 2025), a BO-enhanced RL method; BCEA (Li et al., 2025c), a behavior-centric evolutionary algorithm (EA); and a GRPO (Shao et al., 2024) adaptation with signed standardized advantages. CoPro directly leverages numerical objective/constraint feedback for within-group comparisons and does not require reward shaping; for RL baselines that rely on scalar rewards, we follow the reward specification in rGNN-RL (Li & Carusone, 2023). We use the same uniform scalarization weights for these reward-based baselines across circuits and do not tune weights per task. For all IC design experiments, we run each method 10 times and report the mean performance under a fixed budget of 4000 SPICE evaluations; additional implementation details are provided in Appendix G.

Table 1 shows that CoPro delivers consistently strong performance across eight AnalogGym tasks and objectives: it achieves top-2 results on $\text{FOM}_L$ for all tasks while remaining highly competitive on $\text{FOM}_S$, and it often produces compact designs with a small AREA. Compared with strong baselines and GRPO-style signed reweighting, CoPro im-

*Table 1.* Comparison of different algorithms over eight benchmarks in AnalogGym across 10 independent runs. Red indicates the best result, and blue indicates the second-best result.

| | Algorithm | Metric | NMCF | NMCNR | ACBC | AFFC | DFCFC1 | DFCFC2 | PFC | RAFFC |
|---|---|---|---|---|---|---|---|---|---|---|
| **Heuristic** | Ckt-GNN | $FOM_L \uparrow$ | 236.2 | 179.9 | 699.4 | 359.2 | 502.8 | 515.9 | 185.1 | 723.4 |
| | | $FOM_S \uparrow$ | 617.7 | 923.2 | 1130.8 | 851.5 | 1190.6 | 1526 | 868.4 | 1245.1 |
| | | Area $\downarrow$ | 135.8 | 294.6 | 125.1 | 155.5 | 91.5 | 97.8 | 131.6 | 173.9 |
| | BCEA | $FOM_L \uparrow$ | 367.7 | 269 | 1104.5 | 472 | 772.6 | 871.5 | 280.0 | 847.8 |
| | | $FOM_S \uparrow$ | 924.4 | 987.5 | 1749.5 | 1247.2 | 2050.8 | 2713.6 | 655.3 | 1685.3 |
| | | Area $\downarrow$ | 125.2 | 246.5 | 127.2 | 136.1 | 75.6 | 85.3 | 135.2 | 163.2 |
| **RL-based** | rGNN-RL | $FOM_L \uparrow$ | 162.6 | 115.4 | 407.2 | 275.6 | 330.6 | 277.5 | 126.2 | 603.2 |
| | | $FOM_S \uparrow$ | 407.7 | 847.4 | 699.6 | 589.8 | 749.2 | 628.7 | 293.4 | 913.5 |
| | | Area $\downarrow$ | 142.5 | 330.9 | 126.7 | 171.4 | 104.3 | 112.7 | 155.6 | 176.3 |
| | MA-Opt | $FOM_L \uparrow$ | 253.1 | 298.1 | 788.2 | 382 | 556.7 | 583.8 | 200.4 | 726 |
| | | $FOM_S \uparrow$ | 649.8 | 943.8 | 1264.9 | 925 | 1334.2 | 1762.5 | 611 | 1317 |
| | | Area $\downarrow$ | 131.9 | 281.2 | 125 | 153.8 | 90.2 | 96.1 | 142.5 | 170.9 |
| | MARL | $FOM_L \uparrow$ | 289.4 | 225.6 | 929.1 | 419.1 | 646.5 | 708.1 | 226.6 | 785.2 |
| | | $FOM_S \uparrow$ | 789.8 | 981.9 | 1428.5 | 1044.1 | 1533.9 | 3127.3 | 728.4 | 1480.4 |
| | | Area $\downarrow$ | 127 | 267.1 | 128.6 | 144.1 | 83.1 | 87.9 | 140.3 | 169.9 |
| | RoSE-Opt | $FOM_L \uparrow$ | 362.7 | 243.9 | 1234.6 | 423.6 | 712.7 | 785.5 | 247.9 | 822.8 |
| | | $FOM_S \uparrow$ | 820.5 | 1046.1 | 1546.5 | 1104.3 | 1669.8 | 2406.9 | 774 | 1547 |
| | | Area $\downarrow$ | 131.1 | 260.1 | 123.2 | 139.2 | 80.2 | 79.0 | 127.6 | 162.6 |
| | GRPO | $FOM_L \uparrow$ | 274.4 | 215 | 860.1 | 489.5 | 608.6 | 656.9 | 221.7 | 758.5 |
| | | $FOM_S \uparrow$ | 723.2 | 974.6 | 1365.9 | 976.8 | 1448.5 | 1958.2 | 683 | 1419.7 |
| | | Area $\downarrow$ | 130.6 | 272.8 | 124.5 | 148.4 | 85.1 | 91.3 | 143 | 157.1 |
| | **CoPro (Our)** | $FOM_L \uparrow$ | 354.2 | 294.6 | 1226.6 | 485.5 | 837.3 | 976.3 | 296.3 | 926.9 |
| | | $FOM_S \uparrow$ | 1004.4 | 1068.8 | 1933.4 | 1343.3 | 2038.8 | 3102.3 | 987.3 | 1827.2 |
| | | Area $\downarrow$ | 125.8 | 219.4 | 128.3 | 132.8 | 71.4 | 73.4 | 137.5 | 164.7 |

proves $FOM_S$ and $FOM_L$ on nearly all tasks, with evident gains on more challenging settings such as DFCFC2 and PF. Figure 4 displays the learning curves of various algorithms on the PFC. First, Fig. 4 (b) shows that CoPro rapidly suppresses constraint violations and maintains a low VR. This validates our feasibility-anchored moment construction, which prioritizes updates from informative, feasible candidates over high-reward but infeasible ones. Second, Fig. 4 (a) demonstrates steady Pareto-front progress, with HV rising quickly and stabilizing. This confirms that comparative scoring provides a robust multi-objective signal by focusing on within-group relative improvement rather than fragile scalarization. Finally, Fig. 4(c) shows lower and more stable gradient cancellation for CoPro than GRPO, consistent with reduced sign-flip interference under nonnegative reweighting. More experimental results are provided in Appendix H.

### 5.3. Main Results on Tool Calling

We compare our framework on tool-calling tasks against representative baselines spanning distinct paradigms: GRPO (Shao et al., 2024), a group-relative RL baseline; DAPO (Yu et al., 2025), a decoupled-clip and dynamic-sampling group-relative method; GDPO (Liu et al., 2026), a group reward-decoupled normalization method for multi-reward; and LCPO (Aggarwal & Welleck, 2025), an RL approach for controlling reasoning length. Additional im-

*Table 2.* Tool-calling results on BFCL-v3 with Qwen-2.5-Instruct (1.5B/3B). We report average accuracy, format correctness, and average output length under long and short length settings. Red indicates the best result, and blue indicates the strongest baseline.

| | Metric | CoPro | DAPO | GDPO | LCPO | GRPO |
|---|---|---|---|---|---|---|
| **Qwen-2.5 1.5B (Short)** | Avg Acc $\uparrow$ | 31.7% | 29.0% | 30.9% | 29.5% | 28.3% |
| | Fmt Acc $\uparrow$ | 83.1% | 80.6% | 80.2% | 79.2% | 77.6% |
| | Avg Length | 124.6 | 156.2 | 171.3 | 136.9 | 196.8 |
| **Qwen-2.5 1.5B (Long)** | Avg Acc $\uparrow$ | 33.9% | 30.3% | 31.6% | 30.9% | 29.4% |
| | Fmt Acc $\uparrow$ | 83.5% | 81.0% | 80.5% | 79.5% | 78.8% |
| | Avg Length | 468.2 | 509.6 | 527.4 | 441.8 | 603.5 |
| **Qwen-2.5 3B (Short)** | Avg Acc $\uparrow$ | 40.8% | 38.0% | 39.4% | 38.6% | 37.2% |
| | Fmt Acc $\uparrow$ | 83.7% | 82.3% | 81.9% | 81.0% | 80.4% |
| | Avg Length | 126.3 | 158.5 | 173.8 | 139.1 | 198.4 |
| **Qwen-2.5 3B (Long)** | Avg Acc $\uparrow$ | 42.6% | 39.0% | 40.5% | 39.7% | 38.1% |
| | Fmt Acc $\uparrow$ | 84.9% | 82.6% | 83.1% | 81.9% | 81.1% |
| | Avg Length | 472.5 | 520.4 | 536.2 | 448.7 | 615.9 |

plementation details are provided in Appendix I. Table 2 demonstrates that CoPro achieves the highest average accuracy, optimal format compliance, and shorter outputs across Qwen-2.5 1.5B and 3B models. Among baselines, GDPO is typically the strongest on average accuracy, while LCPO better controls length. CoPro's advantages in format control and length reduction are more pronounced than its gains in average accuracy, indicating that its performance improvement stems primarily from reliable constraint satisfaction. Short-length configurations pose greater challenges for all methods, as tighter length constraints narrow the margin for error correction and increase ambiguity in tool argument

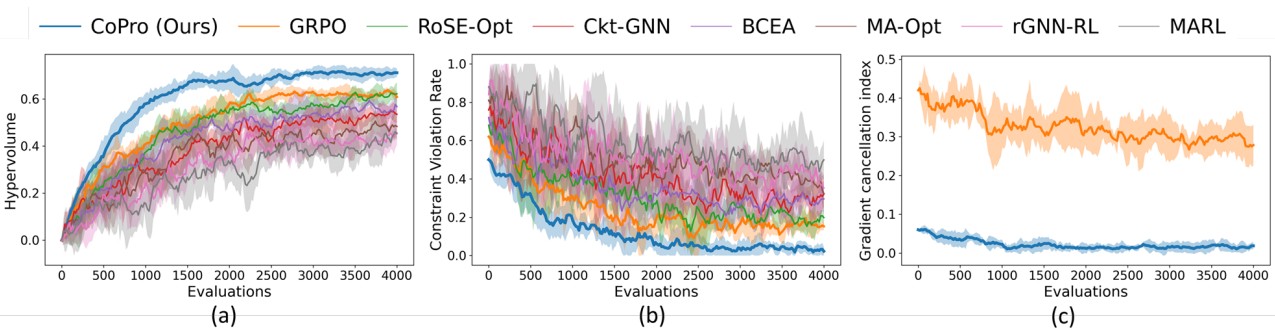

*Figure 4.* Case study on PFC. (a) HV, (b) Constraint VR, and (c) GCI comparison of CoPro and GRPO. Shaded regions indicate variability across runs.

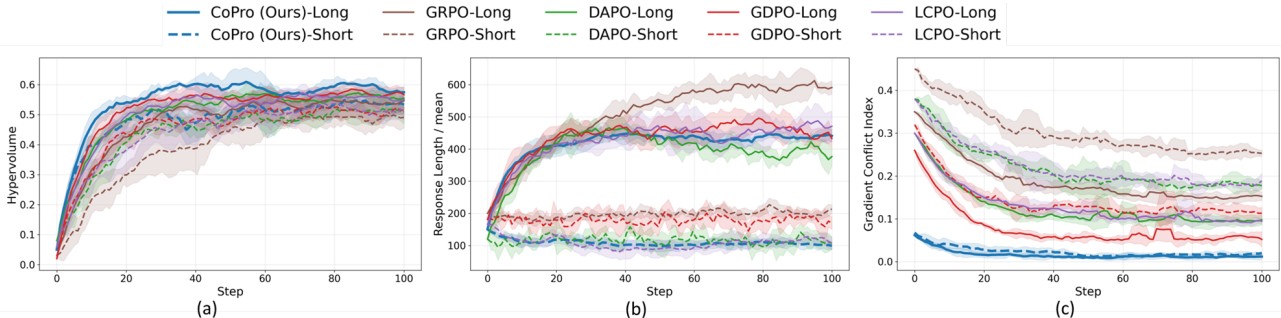

*Figure 5.* Tool-calling results on Qwen-2.5-Instruct 1.5B. (a) HV, (b) response length, and (c) GCI comparison under long/short length settings. Shaded regions indicate variability across runs.

formatting. Because CoPro treats length as a feasibility constraint while reward-based baselines use shaped length rewards, Appendix M also reports a hard-length-constraint comparison for all methods to isolate this design choice.

Figure 5 explains these outcomes via training dynamics. In Fig. 5 (a), CoPro achieves a steeper early HV gain under both long and short settings, indicating that its group-level projection yields more consistently improving updates when correctness and format rewards are mixed. Fig. 5 (b) highlights a key failure mode of reward-based length control: GRPO drifts toward overly long outputs in the long setting, and shows larger length variance even under the short constraint, whereas CoPro maintains a stable length regime without over-generating. This matters for tool calling because verbosity and truncation interact with a brittle, token-level format checker: small deviations in tool/argument tokens can flip the binary format reward, making signed, normalized group advantages noisy and easy to "game" via length. Consistently, Fig. 5 (c) shows CoPro sustains a much lower gradient conflict index throughout training, suggesting fewer sign-flip-induced cancellations on shared tool schemas and a more aligned update direction when correctness, format, and length objectives compete.

*Table 3.* Key ablations on PFC (AnalogGym). Mean±std over 10 runs under 4000 SPICE evaluations. "w/o feasibility moment" removes the E-step feasibility constraint (and the associated archive-anchor injection / relaxation used under scarce feasibility).

| Method | HV ↑ | VR ↓ | GCI ↓ |
|---|---|---|---|
| **CoPro (full)** | **0.683±0.012** | **0.058±0.015** | **0.021±0.008** |
| w/o feasibility moment | 0.550 ± 0.020 | 0.260 ± 0.030 | 0.060 ± 0.020 |
| signed weights | 0.520 ± 0.025 | 0.200 ± 0.030 | 0.450 ± 0.060 |
| w/o distinctiveness moment | 0.600 ± 0.020 | 0.140 ± 0.020 | 0.100 ± 0.030 |
| w/o dense fallback | 0.570 ± 0.020 | 0.170 ± 0.020 | 0.070 ± 0.020 |

### 5.4. Ablations

Table 3 isolates the contributions of CoPro components on PFC. Removing the feasibility moment causes the largest VR increase and a consistent HV drop. This ablation disables the E-step feasibility moment constraint (Sec. 4.2), and therefore also removes the archive-anchor injection mechanism. The resulting update can be dominated by infeasible-but-high-scoring samples, wasting a substantial fraction of the fixed SPICE budget on violating designs (high VR). The HV loss appears more modest because HV is computed on the feasible nondominated subset: once feasibility is recovered later in training, the method can still improve objectives on the remaining feasible set, partially masking early feasibility failures in the final HV summary. Switching from CoPro's nonnegative target distribution to signed weights

produces a dramatic GCI increase and worse HV/VR. Removing the distinctiveness moment further increases GCI and VR, highlighting the importance of down-weighting near-ties. Finally, removing the dense fallback degrades both HV and VR, suggesting that sparse PF-improvement events alone are insufficient to sustain learning and motivating the design of coupling sparse front-expansion with a dense comparative signal in the score. Additional ablation analyses, including direct residual-design ablations, and implementation details are provided in Appendix L and Appendix M.

## 6. Conclusion

CoPro addresses constrained multi-objective learning from within-group comparisons by replacing signed, normalized advantages with a KL-regularized, moment-constrained distribution projection. The resulting nonnegative target distribution preserves feasible anchors and suppresses near-ties, leading to a stable projection-based policy update. Empirically, CoPro improves feasible Pareto-front quality, constraint satisfaction, and optimization stability on analog circuit design and tool-calling benchmarks. This work instantiates the projection with bounded per-sample statistics derived from PF progress, feasibility residuals, and within-group distinctiveness. The main trade-off is that CoPro solves a small convex dual for every sampled group, adding overhead relative to simple standardized-advantage updates such as GRPO. Its behavior also depends on the design of the bounded statistics $(s, g, h)$, although our ablations show that the method is robust to several residual and threshold choices. Future work includes learning these statistics and constraint thresholds from data, reducing projection overhead, and extending the projection to richer constraint sets and longer-horizon settings.

## Acknowledgements

The project was supported by the National Natural Science Foundation of China (Grant No. W2431044).

## Impact Statement

This paper presents work whose goal is to advance the field of Machine Learning. There are many potential societal consequences of our work, none which we feel must be specifically highlighted here.

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

# A. Implementation Instantiation of $s$, Feasibility Residual $g$, and Distinctiveness $h$

This section specifies the bounded statistics $(s_i, g_i, h_i)$ used to instantiate CoPro in our experiments.

**Score construction and squashing.** In Analog IC design (AnalogGym), each candidate $x$ is a circuit design evaluated into a multi-objective vector and constraint feedback. We maintain a feasible PF archive $\mathcal{A}$ in objective space and define a sparse PF-progress term $\Delta\mathrm{HV}(x \mid \mathcal{A})$ as the *feasible* hypervolume improvement induced by inserting $x$ into $\mathcal{A}$ (set to 0 if $x$ is infeasible). We also compute a dense objective surrogate $U_{\mathrm{obj}}(x)$ from normalized objectives (e.g., via a Tchebycheff scalarization) and optionally apply a bounded excess-cost adjustment $u_{\mathrm{excess}}(x)$ (Appendix G). The raw comparative score in AnalogGym is

$$s_{\mathrm{raw}}(x) = \lambda_{\mathrm{hv}} \, \Delta\mathrm{HV}(x \mid \mathcal{A}) + \lambda_u \, U_{\mathrm{obj}}(x) + u_{\mathrm{excess}}(x). \tag{24}$$

In tool calling, each candidate $x$ is a model response and is evaluated into two bounded objective coordinates (used for PF progress): format score $f(x) \in [0, 1]$ and correctness score $c(x) \in [0, 1]$ (clipped to $[0, 1]$). We maintain a feasible PF archive $\mathcal{A}$ in the $(f, c)$ plane and construct a raw comparative score

$$s_{\mathrm{raw}}(x) = \lambda_{\mathrm{hv}} \, \Delta\mathrm{HV}(x \mid \mathcal{A}) + \lambda_u \, m_{\mathrm{dom}}(x) + \lambda_{\mathrm{obj}} \, U_{\mathrm{obj}}(x), \tag{25}$$

where $\Delta\mathrm{HV}(x \mid \mathcal{A})$ denotes the *feasible* hypervolume improvement induced by inserting $(f(x), c(x))$ into the current feasible archive $\mathcal{A}$ (set to 0 if $x$ is infeasible; coordinates are clipped to $[0, 1]$ and we use reference point $(0, 0)$), $m_{\mathrm{dom}}(x) \in [-1, 1]$ is a bounded within-group dominance margin, and $U_{\mathrm{obj}}(x) \in [0, 1]$ is a dense objective surrogate. Concretely, for a prompt-group $\{x_i\}_{i=1}^{G}$ and Pareto dominance in $(f, c)$ space defined by $(f, c) \succ (f', c') \iff (f \geq f' \wedge c \geq c') \wedge (f > f' \vee c > c')$, we compute

$$m_{\mathrm{dom}}(x_i) := \frac{d^-(x_i) - d^+(x_i)}{\max(1, G-1)}, \tag{26}$$

$$d^-(x_i) := \sum_{j \neq i} \mathbf{1}[(f(x_j), c(x_j)) \succ (f(x_i), c(x_i))], \tag{27}$$

$$d^+(x_i) := \sum_{j \neq i} \mathbf{1}[(f(x_i), c(x_i)) \succ (f(x_j), c(x_j))]. \tag{28}$$

The dense surrogate is instantiated as a weighted distance-to-$(1, 1)$ score:

$$U_{\mathrm{obj}}(x) := 1 - \frac{\sqrt{w_f^2(1 - f(x))^2 + w_c^2(1 - c(x))^2}}{\sqrt{w_f^2 + w_c^2}}. \tag{29}$$

We then map $s_{\mathrm{raw}}$ to a bounded statistic

$$s(x) = \mathrm{tanh\_normalize}\big(s_{\mathrm{raw}}(x)\big) \in [-1, 1], \tag{30}$$

where $\mathrm{tanh\_normalize}(\cdot)$ is computed *within the group* by standardizing $s_{\mathrm{raw}}$ using the group mean and standard deviation, followed by a temperature-scaled $\tanh$.

**Feasibility residual from violation.** Let $v(x) \geq 0$ denote the aggregated constraint violation with $v(x) = 0$ indicating feasibility. In our tool-calling setting, $v(x)$ is constructed from a length-related violation and (optionally) a format-related violation:

$$v(x) := v_{\mathrm{len}}(x) + v_{\mathrm{fmt}}(x), \qquad v_{\mathrm{fmt}}(x) := \mathrm{scale}_{\mathrm{fmt}} \cdot (1 - f(x)), \tag{31}$$

where $v_{\mathrm{fmt}}$ can be enabled to treat format as an explicit constraint (otherwise $v_{\mathrm{fmt}}(x) = 0$). The length signal may be provided either as a nonnegative violation or as a residual (where $\leq 0$ indicates satisfaction), and we convert it into $v_{\mathrm{len}}(x) \geq 0$ accordingly. In AnalogGym, $v(x)$ aggregates hard constraint violations (and we optionally track an excess cost separately), and we set $\mathrm{scale}_g$ to a robust running estimate of typical violation magnitudes (e.g., an EMA of the median infeasible violation, with clipping) for numerical stability. We map violations to a signed feasibility residual by

$$g(x) = \begin{cases} -\varepsilon_{\mathrm{feas}}, & \text{if } v(x) = 0, \\ \min\big\{g_{\mathrm{max}}, \log\big(1 + v(x)/\mathrm{scale}_g\big)\big\}, & \text{if } v(x) > 0, \end{cases} \tag{32}$$

where $\varepsilon_{\text{feas}} > 0$, $\text{scale}_g > 0$, and $g_{\max} > 0$ are constants. This construction ensures feasible designs form strict negative anchors while infeasible designs obtain bounded positive residuals that grow monotonically with the degree of violation before clipping.

**Distinctiveness from within-group score separation.** Given a group of scores $\{s(x_i)\}_{i=1}^G$, we define distinctiveness as

$$h(x_i) = h_{\max}\Big(1 - \exp\big(-|s(x_i) - \text{median}(\{s(x_j)\}_{j=1}^G)|/\text{temp}\big)\Big), \tag{33}$$

where $h_{\max} > 0$ and $\text{temp} > 0$. This mapping assigns small $h$ to near-ties around the group median and increases $h$ with comparative separation.

## B. Additional Proofs and KKT/Dual Details

### B.1. Detailed Proof of Lemma 4.2

Recall $\mathcal{F} = \{i : g_i \leq 0\}$ and $\mathcal{I} = \{1, \ldots, G\} \setminus \mathcal{F}$ with $g_{\min}^- = \min_{i \in \mathcal{F}} g_i < 0$ and $g_{\min}^+ = \min_{i \in \mathcal{I}} g_i > 0$. Let $q(\mathcal{F}) = \sum_{i \in \mathcal{F}} q_i \in [0, 1]$.

We prove the lower bound by considering the optimization problem that produces the *smallest feasible-anchor mass*:

$$\min_{q \in \Delta^G} q(\mathcal{F}) \quad \text{s.t.} \quad \sum_{i=1}^G q_i g_i \leq 0. \tag{34}$$

**Step 1: Reduction to extreme points within each subset.** Fix any feasible $q$ and denote $q_F := q(\mathcal{F})$. Define $i^- \in \arg\min_{i \in \mathcal{F}} g_i$ and $i^+ \in \arg\min_{i \in \mathcal{I}} g_i$. Construct a new distribution $\tilde{q}$ that preserves the *same* total mass $q_F$ on $\mathcal{F}$ and $1 - q_F$ on $\mathcal{I}$, but concentrates it on the most favorable residuals:

$$\tilde{q}_{i^-} = q_F, \quad \tilde{q}_{i^+} = 1 - q_F, \quad \tilde{q}_i = 0 \text{ otherwise.}$$

Then

$$\sum_{i=1}^G \tilde{q}_i g_i = q_F g_{\min}^- + (1 - q_F) g_{\min}^+ \leq \sum_{i=1}^G q_i g_i, \tag{35}$$

because (i) for $i \in \mathcal{F}$ we have $g_i \geq g_{\min}^-$ and concentrating mass on $g_{\min}^-$ can only decrease the feasible contribution, and (ii) for $i \in \mathcal{I}$ we have $g_i \geq g_{\min}^+$ and concentrating mass on $g_{\min}^+$ can only decrease the infeasible contribution. Therefore, if $q$ satisfies $\sum_i q_i g_i \leq 0$, then $\tilde{q}$ with the same $q_F$ also satisfies it. Hence, the minimum in (34) is attained by a two-point distribution supported on $\{i^-, i^+\}$.

**Step 2: Solve the reduced inequality.** For such $\tilde{q}$, the feasibility constraint becomes

$$q_F g_{\min}^- + (1 - q_F) g_{\min}^+ \leq 0. \tag{36}$$

Rearranging gives

$$q_F(g_{\min}^+ - g_{\min}^-) \geq g_{\min}^+ \quad \Rightarrow \quad q_F \geq \frac{g_{\min}^+}{g_{\min}^+ - g_{\min}^-}.$$

Since $g_{\min}^+ > 0$ and $g_{\min}^- < 0$, the denominator is strictly positive and the bound is in $(0, 1)$. This proves (15).

**Remark (tightness).** The bound is tight: equality holds for the two-point distribution supported on $i^-$ and $i^+$ that satisfies (36) with equality.

### B.2. KKT Derivation and Numerical Dual Solver for (12)

This section derives the closed form (14) and provides a lightweight solver for the dual variables $(\kappa, \mu)$.

### B.2.1. LAGRANGIAN AND CLOSED-FORM $q^*$

We consider the KL-regularized E-step (Sec. 4.2):

$$\max_{q \in \Delta^G} \sum_i q_i s_i - \tau_{\mathrm{KL}} \mathrm{KL}(q\|u) \quad \text{s.t.} \quad \sum_i q_i g_i \leq 0, \quad \sum_i q_i h_i \geq \tau_c,$$

with $\tau_{\mathrm{KL}} > 0$. Introduce multipliers $\kappa \geq 0$ and $\mu \geq 0$ for $\sum_i q_i g_i \leq 0$ and $\tau_c - \sum_i q_i h_i \leq 0$, and $\lambda \in \mathbb{R}$ for $\sum_i q_i = 1$. The Lagrangian is

$$\mathcal{L}(q, \kappa, \mu, \lambda) = \sum_i q_i s_i - \tau_{\mathrm{KL}} \sum_i q_i \log \frac{q_i}{u_i} - \kappa \sum_i q_i g_i + \mu \Big( \sum_i q_i h_i - \tau_c \Big) + \lambda \Big( \sum_i q_i - 1 \Big). \tag{37}$$

Stationarity w.r.t. $q_i$ yields

$$\frac{\partial \mathcal{L}}{\partial q_i} = s_i - \tau_{\mathrm{KL}} \Big( 1 + \log \frac{q_i}{u_i} \Big) - \kappa g_i + \mu h_i + \lambda = 0,$$

hence

$$q_i = u_i \exp \Big( \frac{s_i - \kappa g_i + \mu h_i + \lambda - \tau_{\mathrm{KL}}}{\tau_{\mathrm{KL}}} \Big) \propto u_i \exp \Big( \frac{s_i - \kappa g_i + \mu h_i}{\tau_{\mathrm{KL}}} \Big),$$

and normalization over $\Delta^G$ gives (14).

### B.2.2. CONVEX DUAL OBJECTIVE AND GRADIENTS

Define

$$Z(\kappa, \mu) := \sum_{i=1}^{G} u_i \exp \Big( \frac{s_i - \kappa g_i + \mu h_i}{\tau_{\mathrm{KL}}} \Big), \qquad q_i(\kappa, \mu) := \frac{u_i \exp \Big( \frac{s_i - \kappa g_i + \mu h_i}{\tau_{\mathrm{KL}}} \Big)}{Z(\kappa, \mu)}. \tag{38}$$

Maximizing (37) over $q \in \Delta^G$ yields the dual function

$$\mathcal{D}(\kappa, \mu) = \tau_{\mathrm{KL}} \log Z(\kappa, \mu) - \mu \tau_c, \qquad \kappa \geq 0, \ \mu \geq 0, \tag{39}$$

and the dual problem is $\min_{\kappa, \mu \geq 0} \mathcal{D}(\kappa, \mu)$. This objective is convex because $\log Z$ is a log-sum-exp.

Let $\mathbb{E}_q[\cdot]$ denote expectation under $q(\kappa, \mu)$. Then the gradients are

$$\frac{\partial \mathcal{D}}{\partial \kappa} = -\mathbb{E}_q[g], \qquad \frac{\partial \mathcal{D}}{\partial \mu} = \mathbb{E}_q[h] - \tau_c. \tag{40}$$

Thus a projected gradient step for minimizing (39) is

$$\kappa \leftarrow \big[ \kappa + \alpha \, \mathbb{E}_q[g] \big]_+, \qquad \mu \leftarrow \big[ \mu + \alpha \, (\tau_c - \mathbb{E}_q[h]) \big]_+. \tag{41}$$

**Newton step.** The Hessian entries can be written via covariances under $q$:

$$\frac{\partial^2 \mathcal{D}}{\partial \kappa^2} = \frac{1}{\tau_{\mathrm{KL}}} \mathrm{Var}_q(g), \qquad \frac{\partial^2 \mathcal{D}}{\partial \mu^2} = \frac{1}{\tau_{\mathrm{KL}}} \mathrm{Var}_q(h), \qquad \frac{\partial^2 \mathcal{D}}{\partial \kappa \partial \mu} = -\frac{1}{\tau_{\mathrm{KL}}} \mathrm{Cov}_q(g, h). \tag{42}$$

Solve $(H + \lambda I)\Delta = \nabla \mathcal{D}$ with small damping $\lambda > 0$ and update $(\kappa, \mu) \leftarrow \Pi_{\mathbb{R}_+^2} \big( (\kappa, \mu) - \Delta \big)$.

**Numerical notes.** We compute logits $\ell_i = (s_i - \kappa g_i + \mu h_i)/\tau_{\mathrm{KL}} + \log u_i$ and use a stable log-sum-exp: $q_i \propto \exp(\ell_i - \max_j \ell_j)$. With warm starts from the previous group, a few projected steps (41) (or 1–3 Newton steps) are typically sufficient since $G$ is small.

### B.2.3. SMOOTHED DUAL UPDATES

In noisy or nonstationary settings, we smooth the per-group dual gradients via an exponential moving average (EMA). Let $\widehat{g}_t := \mathbb{E}_{q_t}[g]$ and $\widehat{h}_t := \mathbb{E}_{q_t}[h]$ for group $t$, and define EMA states $m_t^g = (1 - \beta_{\mathrm{ema}})m_{t-1}^g + \beta_{\mathrm{ema}} \widehat{g}_t$ and $m_t^h = (1 - \beta_{\mathrm{ema}})m_{t-1}^h + \beta_{\mathrm{ema}}(\tau_c - \widehat{h}_t)$. A smoothed projected update is

$$\kappa \leftarrow \big[ \kappa + \alpha \, m_t^g \big]_+, \qquad \mu \leftarrow \big[ \mu + \alpha \, m_t^h \big]_+, \tag{43}$$

which reduces to (41) when $\beta_{\mathrm{ema}} = 1$.

### B.3. Feasibility Fallback and Relaxation (Formal Statements)

This section formalizes the two fallback mechanisms mentioned in Sec. 4.2: anchor injection and relaxed feasibility moments.

**Proposition B.1** (Anchor injection guarantees feasibility of the E-step). *Assume the archive contains at least one evaluated feasible design $x_{\mathrm{anc}}$ with feasibility residual $g_{\mathrm{anc}} = -\varepsilon_{\mathrm{feas}} < 0$ (cf. (32)). Consider augmenting a group by adding $x_{\mathrm{anc}}$ (no additional environment queries) and solving the E-step on the augmented set. Then the feasibility constraint $\sum_i q_i g_i \leq 0$ is feasible for the augmented problem.*

**Proof.**  Let $g_{\min}^+$ be the minimum infeasible residual among the (original) group; if all original samples are infeasible then $g_{\min}^+ > 0$. Consider a distribution supported only on $\{x_{\mathrm{anc}}, i^+\}$, where $i^+$ attains $g_{\min}^+$: set $q_{\mathrm{anc}} = \frac{g_{\min}^+}{g_{\min}^+ + \varepsilon_{\mathrm{feas}}}$ and $q_{i^+} = 1 - q_{\mathrm{anc}}$. Then $q_{\mathrm{anc}}(-\varepsilon_{\mathrm{feas}}) + (1 - q_{\mathrm{anc}})g_{\min}^+ = 0$, so $\sum_i q_i g_i \leq 0$ holds. Hence the feasibility moment constraint is satisfiable on the augmented group. □

**Distinctiveness feasibility.**  With the adaptive choice $\tau_c = \rho \max_i h_i$ (Sec. 4.2), the constraint $\sum_i q_i h_i \geq \tau_c$ is feasible whenever at least one $h_i > 0$; when all $h_i = 0$ we set $\tau_c = 0$ and the constraint is vacuous. Thus, anchor injection does not introduce infeasibility for the distinctiveness constraint.

**Proposition B.2** (Relaxed feasibility moment ensures a well-defined E-step without feasible anchors). *If no feasible anchor exists (e.g., the archive is empty early in training), consider the relaxed E-step obtained by replacing $\sum_i q_i g_i \leq 0$ with $\sum_i q_i g_i \leq \delta_t$ for some $\delta_t \geq 0$. If $\delta_t \geq \min_i g_i$ for the current group, then the relaxed constraint is feasible. Moreover, choosing any schedule $\delta_t \downarrow 0$ yields a sequence of well-defined E-steps; once a feasible sample (or an injected feasible anchor) becomes available, we set $\delta_t = 0$ and recover the original constraint.*

**Proof.**  Since $\sum_i q_i g_i$ is a convex combination, $\sum_i q_i g_i \geq \min_i g_i$ for all $q \in \Delta^G$. Therefore, if $\delta_t \geq \min_i g_i$, there exists at least one feasible $q$ (e.g., the point mass on $\arg\min_i g_i$) satisfying $\sum_i q_i g_i \leq \delta_t$. The remaining statements follow by construction of the schedule and switching back to $\delta_t = 0$ once feasibility is available. □

**Practical instantiation.**  A simple choice is $\delta_t = \delta_0 \exp(-t/T)$ with $\delta_0$ set to a robust upper bound on early violations. We keep $\delta_t = +\infty$ until the first feasible point appears, then set $\delta_t = 0$ and rely on Proposition B.1 thereafter.

## C. Connection to KL-Regularized Policy Search

The E-step in CoPro is closely related to KL-regularized policy search and mirror-descent-style updates, including relative-entropy policy search (REPS) (Peters et al., 2010) and maximum a posteriori policy optimization (MPO) (Abdolmaleki et al., 2018). This perspective also connects to trust-region KL-constrained policy updates (TRPO) (Schulman et al., 2015), the theory of regularized Markov decision processes (Geist et al., 2019), and mirror-descent unifications of policy optimization (Tomar et al., 2022). These methods compute an improved nonparametric distribution by optimizing expected utility under a KL constraint (or, equivalently, maximizing utility minus a KL penalty), yielding an exponential-family reweighting over samples; the parametric policy is then projected toward that target distribution. Our formulation follows the same high-level "improve in distribution space, then project" template, but applies it at the group level using bounded comparative statistics and moment constraints that encode feasibility and discourage near-ties. Algorithm 1 provides an end-to-end view of this "distribution improvement → projection" loop in our setting.

## D. Connection to Group-Relative Optimization

GRPO was developed in the context of large-model post-training as a practical, critic-free alternative to value-function-based policy gradients. Its core idea is to sample a group of outputs under the same conditioning context, construct within-group relative advantages (typically standardized by the group mean and variance), and apply PPO-style updates with clipping and KL regularization (Shao et al., 2024; Schulman et al., 2017). GRPO has been adopted broadly in large-scale post-training and RLVR-style pipelines (Guo et al., 2025; Zhang & Zuo, 2025), motivating a growing set of practical refinements. On the stability side, prior work studies decoupled clipping and dynamic sampling for long-horizon generation (Yu et al., 2025; Liu et al., 2025a), analyzes how clipping choices shift entropy and contribute to entropy collapse (Park et al., 2025), and proposes controllable-entropy regularization frameworks (Wang et al., 2025). On the objective side, recent work revisits

GRPO through data-efficiency and preference/contrastive-learning perspectives (Wu et al., 2025), and develops efficiency and correction mechanisms such as completion pruning and dynamic allocation (Lin et al., 2025) and trajectory-corrected updates (Pang et al., 2025).

In this paper we adopt the same within-group sampling and relative-learning paradigm, but the comparison signal is induced by constrained multi-objective feedback rather than a single scalar reward. Under feasibility scarcity and Pareto near-ties, mapping these structured comparisons to signed, normalized scalar weights can be brittle, motivating our distribution-projection view in which the group is reweighted by a nonnegative target distribution before projecting the policy. Algorithm 1 summarizes the concrete training loop used throughout the paper.

## E. Gradient Cancellation, Entropy, and Pareto-Front Progress

This section formalizes the connection between (i) sign-flip gradient cancellation, (ii) concentration/entropy of the E-step weights, and (iii) stable progress on the feasible PF.

**Assumption E.1** (Shared component and independent residuals). For a fixed context $c$ and group $\{x_i\}_{i=1}^G$, the score-function features satisfy

$$\phi_i = \nabla_\theta \log \pi_\theta(x_i \mid c) = u + \varepsilon_i, \qquad \mathbb{E}[\varepsilon_i] = 0, \qquad \mathbb{E}[\varepsilon_i \varepsilon_i^\top] = \sigma^2 I, \tag{44}$$

where $u$ captures a shared update direction induced by reusable patterns and $\{\varepsilon_i\}$ are independent across $i$.

**Sign flips attenuate the shared direction.** Consider an aggregated update $g(a) = \sum_{i=1}^G a_i \phi_i$ with weights $a_i = \alpha_i s_i$, where $\alpha_i \geq 0$ and $s_i \in \{+1, -1\}$ is a random sign (e.g., induced by signed, normalized group advantages). Let $p = \mathbb{P}(s_i = -1)$ and assume $\{s_i\}$ are independent across $i$ with $\mathbb{E}[s_i] = 1 - 2p$.

**Proposition E.2** (Shared-component retention under sign flips). *Under Assumption E.1, let $\alpha \in \Delta^G$ be nonnegative weights with $\sum_i \alpha_i = 1$ and define $S := \sum_i \alpha_i s_i$. Then*

$$\mathbb{E}[S^2] = (1 - 2p)^2 + \left(1 - (1 - 2p)^2\right) \sum_{i=1}^G \alpha_i^2 = (1 - 2p)^2 + \frac{1 - (1 - 2p)^2}{\text{ESS}(\alpha)}, \tag{45}$$

*and the shared-component contribution to the squared update magnitude satisfies*

$$\mathbb{E}\left[\left\|\sum_{i=1}^G \alpha_i s_i \, u\right\|^2\right] = \|u\|^2 \, \mathbb{E}[S^2]. \tag{46}$$

*In contrast, for nonnegative aggregation (no sign flips), $S \equiv 1$ and the shared term equals $\|u\|^2$.*

Since $s_i^2 = 1$ and $s_i$ are independent, $\mathbb{E}[s_i s_j] = \mathbb{E}[s_i]\mathbb{E}[s_j] = (1 - 2p)^2$ for $i \neq j$. Therefore, $\mathbb{E}[S^2] = \sum_i \alpha_i^2 + \sum_{i \neq j} \alpha_i \alpha_j (1 - 2p)^2 = (1 - 2p)^2 (\sum_i \alpha_i)^2 + \left(1 - (1 - 2p)^2\right) \sum_i \alpha_i^2$. Substituting $\sum_i \alpha_i = 1$ and $\text{ESS}(\alpha) = 1 / \sum_i \alpha_i^2$ gives (45), and (46) follows by linearity.

Proposition E.2 identifies a concrete mechanism: when comparisons are ambiguous (near-ties), $p$ approaches $1/2$ and $(1 - 2p)^2$ collapses, so signed aggregation loses the shared direction $u$ that must accumulate across iterations for sustained improvement. The dependence on $\sum_i \alpha_i^2$ also shows that concentration (low ESS) interacts with sign flips. Fig. 6 visualizes (45) and the geometry of aggregation.

**From cancellation to PF progress.** Feasible PF progress requires that updates accumulate coherently along reusable directions that improve feasibility and objective trade-offs across contexts. Under Assumption E.1, this is captured by the shared component $u$. Proposition E.2 shows that signed aggregation reduces the effective contribution of $u$ precisely in the regime where comparisons are ambiguous (near-ties, noisy constraints), disrupting accumulation across iterations and reducing sample efficiency. CoPro's E-step avoids sign flips by optimizing a *nonnegative* reweighting $q^* \in \Delta^G$ and then projecting the policy toward $q^*$, making the shared component additive across samples that share patterns.

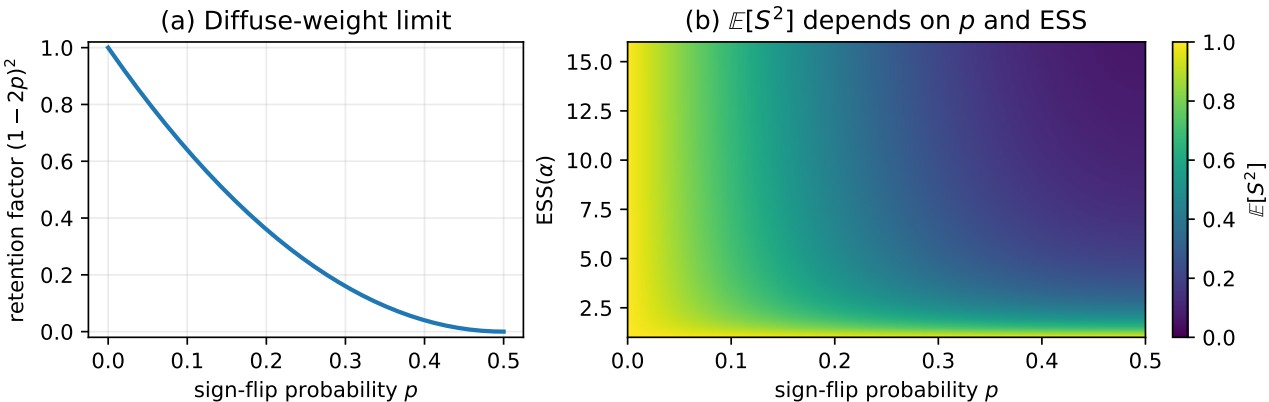

*Figure 6.* Gradient cancellation and sign flips. Left: the diffuse-weight limit $(1-2p)^2$ in (45). Right: $\mathbb{E}[S^2]$ in (45) as a function of sign-flip probability $p$ and $\mathrm{ESS}(\alpha)$, showing the interaction between sign flips and weight concentration.

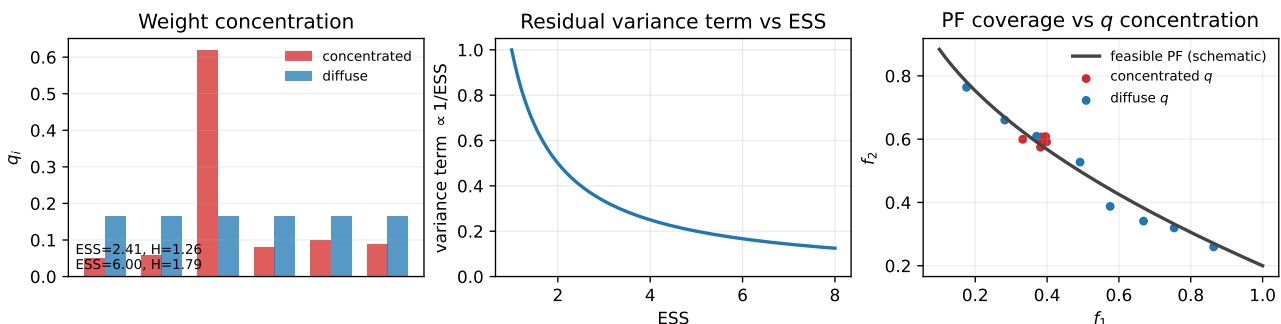

*Figure 7.* Weight concentration and stability. Left: examples of concentrated vs. diffuse weights and their ESS/entropy. Middle: the residual variance term in (47) decreases with ESS. Right: diffuse weights support broader coverage along a feasible PF, while concentrated weights cluster.

**ESS/entropy controls the residual variance.** Let $\alpha \in \Delta^G$ be nonnegative weights and define the residual contribution $r(\alpha) := \sum_i \alpha_i \varepsilon_i$. Under Assumption E.1,

$$\mathbb{E}\big[\|r(\alpha)\|^2\big] = D\sigma^2 \sum_{i=1}^{G} \alpha_i^2 = \frac{D\sigma^2}{\mathrm{ESS}(\alpha)}. \tag{47}$$

where $D$ is the dimension of the score-function feature space. Thus higher ESS (and, typically, higher entropy) reduces the residual-variance term of the update estimator, stabilizing PF expansion by reducing sensitivity to tie-breaking and noisy comparisons. Fig. 7 visualizes the ESS/entropy dependence implied by (47).

## F. Pseudocode

Algorithm 1 details the full CoPro training loop. For each context, we sample a group, evaluate feedback, and construct bounded per-sample statistics $(s, g, h)$ (Appendix A). We then solve the KL-regularized moment projection (Eq. (12)), with $\tau_c$ chosen by the adaptive rule (Eq. (13)); if a group contains no feasible samples, we apply anchor injection or a relaxed feasibility moment (Appendix B.3). The reweighting distribution $q^*$ is obtained via the closed-form solution (Eq. (14)) by minimizing the convex 2D dual (Appendix B.2). Finally, we update the policy by projecting toward $q^*$ via the KL-regularized projection objective (Eq. (17)).

---

**Algorithm 1** CoPro (per-iteration update)

---

**Require:** Old policy $\pi_{\theta_{\mathrm{old}}}$, reference policy $\pi_{\mathrm{ref}} \leftarrow \pi_{\theta_{\mathrm{old}}}$, archive $\mathcal{A}_{PF}$, group size $G$, KL temperature $\tau_{\mathrm{KL}}$, trust-region weight $\beta$, distinctiveness ratio $\rho$.

1: **for** each context $c$ **do**
2:    Sample a group $\{x_i\}_{i=1}^{G} \sim \pi_{\theta_{\mathrm{old}}}(\cdot \mid c)$ and evaluate $y(x_i) = (f(x_i), v(x_i))$.
3:    Compute bounded statistics $(s_i, g_i, h_i)$ from $y(x_i)$ and $\mathcal{A}_{PF}$ (Appendix A).
4:    Set $\tau_c \leftarrow \rho \cdot \max_i h_i$ (or $\tau_c \leftarrow 0$ if $\max_i h_i = 0$).
5:    **if** no feasible anchor in group (all $g_i > 0$) **then**
6:        Inject an archive anchor $x_{\mathrm{anc}} \in \mathcal{A}_{PF}$ (no new queries) and augment $(s, g, h)$; if $\mathcal{A}_{PF} = \emptyset$, use relaxed feasibility moment $\sum_i q_i g_i \leq \delta_t$ (Appendix B.3).
7:    **end if**
8:    Solve the 2D convex dual for $(\kappa, \mu) \in \mathbb{R}_+^2$ (Appendix B.2) and obtain
9:        $q_i^* \propto u_i \exp\left((s_i - \kappa g_i + \mu h_i)/\tau_{\mathrm{KL}}\right)$, normalized over the (augmented) group.
10:    Update $\theta$ by minimizing the projection objective
11:        $\mathbb{E}_c[-\sum_i q_i^* \log \pi_\theta(x_i \mid c) + \beta \, \mathrm{KL}(\pi_\theta(\cdot \mid c) \| \pi_{\mathrm{ref}}(\cdot \mid c))]$.
12:    Update archive $\mathcal{A}_{PF}$ with newly found feasible, nondominated candidates.
13: **end for**

---

*Table 4.* Design-variable ranges for the SkyWater SKY130 (130 nm) node used in AnalogGym.

| Category | Parameter | Range / setting |
|---|---|---|
| Design vars | Length (nm) | 130–5000 |
| Design vars | Width (nm) | 200–10000 |
| Design vars | Multiplier | 1–1000 (integer) |
| Design vars | Input bias current ($\mu$A) | 0.5–100 |

## G. AnalogGym Experimental Details

**Simulation environment.**    We evaluate analog circuit design tasks using AnalogGym (Li et al., 2025a). Circuit performance is obtained by SPICE simulation with Ngspice (Ngspice Development Team, 2025) under the default operating conditions provided by the benchmark. All experiments use the open-source SkyWater 130 nm PDK (Google & Foundry, 2020). We do not introduce additional process/voltage/temperature sweeps beyond the benchmark's default evaluation scripts.

**Design-variable ranges.**    AnalogGym uses the SkyWater SKY130 (130 nm) node, where "130 nm" denotes the technology node of the underlying PDK. The benchmark specifies bounded ranges for design variables (e.g., device widths/lengths and integer multipliers). In our experiments, we follow the benchmark's default operating conditions without introducing additional sweeps. Table 4 summarizes the design-variable ranges used by the 130 nm configuration.

**Objectives and constraints.**    Each AnalogGym task specifies a set of design variables together with a collection of circuit specifications. We optimize three objectives: $\mathrm{FOM}_S$, $\mathrm{FOM}_L$ (Eq. (20)), and AREA; all remaining specifications are treated as constraints and aggregated into a nonnegative violation $v(x) \geq 0$ with $v(x) = 0$ indicating feasibility. For CoPro, we directly use the multi-objective feedback and violation $(f(x), v(x))$ to construct within-group comparisons and the E-step statistics, without defining a scalar reward. For a maximization constraint $m_k(x) \geq t_k$ we use residual $\max(0, (t_k - m_k(x))/(|t_k| + \epsilon))$, and for a minimization constraint $m_k(x) \leq t_k$ we use $\max(0, (m_k(x) - t_k)/(|t_k| + \epsilon))$; $v(x)$ is the sum of these normalized residuals over all constrained specifications. Hypervolume is computed on direction-aware normalized objectives with reference point $r = (0, 0, 0)$, and Table 1 reports the final feasible nondominated archive summary by taking the best coordinate in the reporting direction for each run and averaging over 10 runs.

**Soft vs. hard constraints.**    In our AnalogGym setup, PM (phase margin) is treated as a *hard* constraint that defines feasibility. Other specification metrics are treated as *soft* constraints whose residuals contribute to the aggregate violation $v(x)$. Here, PSRR denotes the power supply rejection ratio, CMRR the common-mode rejection ratio, and GAIN the open-loop gain; $T_s$ denotes settling time.

*Table 5.* AnalogGym training configuration (CoPro).

| Parameter | Value |
|---|---|
| train.num_steps | 500 |
| train.designs_per_step ($G$) | 8 |
| train.SPICE_budget | 4000 evaluations |
| env.max_episode_steps | 50 |
| optim.learning_rate | $1 \times 10^{-4}$ |
| optim.max_grad_norm | 2.0 |
| ppo.epochs | 4 |
| ppo.clip_epsilon | 0.2 |
| algo.target_kl | 0.02 |
| algo.entropy_coef | 0.01 |

**Scalar reward for RL baselines.** For RL baselines that require a scalar reward, we adopt a simple linear shaping based on distances to task targets. Let $\{m_k(x)\}_{k=1}^K$ be the reported specification metrics with target values $\{t_k\}$ and directions (maximize/minimize) given by the task. We define a nonnegative normalized distance

$$d_k(x) = \begin{cases} \max\left(0, \frac{t_k - m_k(x)}{|t_k| + \epsilon}\right), & \text{if } m_k \text{ is to be maximized,} \\ \max\left(0, \frac{m_k(x) - t_k}{|t_k| + \epsilon}\right), & \text{if } m_k \text{ is to be minimized,} \end{cases} \tag{48}$$

and the shaped reward

$$R(x) = -\sum_{k=1}^K d_k(x), \tag{49}$$

where $\epsilon > 0$ is a small constant. This corresponds to treating all specification terms as equally important (i.e., using uniform weights). We do not tune these scalarization weights separately for individual circuits. In contrast, CoPro does not rely on reward design and instead compares candidates directly using objective/constraint feedback.

**Optimization hyperparameters.** We use the same training configuration for all eight AnalogGym circuits. We optimize each circuit under a fixed budget of 4000 SPICE evaluations, implemented as 500 training iterations with 8 sampled designs per iteration. Table 5 summarizes the main hyperparameters used by our AnalogGym runs.

# H. Additional Experimental Comparisons

This section provides additional experimental comparisons to help position CoPro against widely used constraint-handling MORL/RL baselines. In particular, we include five representative baselines: CRPO (Xu et al., 2021), a last-iterate-convergent primal-dual method (LI-PD) (Ding et al., 2023), Exterior Penalty Policy Optimization (EPPO) (Gao et al., 2024), Conflict-Averse Gradient Aggregation (CAGA) (Kim et al., 2025), and Efficient Discovery of Pareto Front (EDPF) (Liu et al., 2025b). These baselines are included primarily to provide a reference cross-comparison with prior constrained MORL literature under analogous constraint settings, complementing the main AnalogGym results.

**Learning curves (HV/VR).** Fig. 8 reports HV and VR learning curves on all eight AnalogGym circuits under the same 4000-evaluation budget.

CoPro consistently drives VR down faster and keeps it lower throughout training, which indicates that a larger fraction of the fixed SPICE budget is spent on *feasible* candidates rather than repeatedly exploring infeasible regions near the constraint boundary. This behavior is a direct consequence of CoPro's E-step feasibility moment constraint $\sum_i q_i g_i \leq 0$ (Sec. 4.2): whenever a group contains feasible (or near-feasible) anchors, the optimal reweighting $q^*$ must assign them strictly positive mass (Lemma 4.2), preventing the update from being dominated by high-scoring but infeasible samples. In contrast, penalty-only or dual-driven baselines can exhibit larger VR bursts when the constraint signal is sparse/noisy under expensive SPICE evaluations.

HV trends are more circuit-dependent rather than preserving a fixed ranking across tasks. When feasibility is scarce and Pareto-relevant improvements are rare (e.g., PFC/DFCFC2), CoPro shows steadier HV gains while maintaining low VR: the nonnegative target distribution reduces sign-flip-induced cancellation on shared design patterns (Sec. 3), and the

distinctiveness moment $\sum_i q_i h_i \geq \tau_c$ suppresses near-ties that otherwise inflate variance under partial orders. On circuits where rapid PF expansion is achievable, PF-discovery-oriented methods can reach higher end-of-budget HV, but typically with looser constraint satisfaction (higher VR) and more oscillatory trajectories; this aligns with CoPro explicitly prioritizing *feasible* PF improvement and using KL-regularized projection to keep updates stable.

**Radar summary.** Fig. 9 summarizes representative circuit-level performance profiles with a radar plot. The radar view highlights *how* each method allocates improvement across heterogeneous specifications (gain/PSRR/CMRR/settling time/area and FOM variants), which is crucial in constrained multi-objective design where a single scalar metric can hide constraint violations or dominance relations. CoPro more often yields balanced profiles across axes, consistent with its comparison-based construction: it reweights candidates by feasible PF progress under explicit feasibility moments, rather than collapsing objectives/constraints into a single signed advantage that can over-emphasize one metric and destabilize updates.

Moreover, the radar plots complement the VR curves by showing that CoPro's lower VR is not achieved by trivially sacrificing objective quality across the board. Instead, the projection step produces a feasibility-aligned update direction while still allowing trade-offs along different objective dimensions to vary by circuit. Overall, taken together with Fig. 8, these per-axis profiles support the intended role of CoPro's projection: robust constraint satisfaction under feasibility scarcity and near-ties, while maintaining competitive multi-objective performance through PF-aligned, nonnegative reweighting.

**Pareto fronts in objective space.** To visualize trade-offs directly in the objective space, Fig. 10 plots the discovered feasible Pareto sets for each circuit using $(\mathrm{FOM}_S, \mathrm{FOM}_L)$. CoPro produces more *feasible* and *well-spread* fronts: the feasibility moment constraint prevents collapsing onto infeasible-but-high-reward regions, while the distinctiveness moment discourages over-concentration on near-ties, leading to broader PF coverage under the same evaluation budget.

# I. Tool-Calling Format

For tool-calling experiments, we follow the data format and verification protocol used in ToolRL-style training (Qian et al., 2025). Each model output is required to contain three fields in the following order:

1. `<think>...</think>`: free-form reasoning;

2. `<tool_call>...</tool_call>`: one or more tool invocations, each written as a JSON object;

3. `<response>...</response>`: the final natural-language answer returned to the user after tool use.

The verifier checks that all fields appear and that their order is correct.

**Tool-call JSON schema.** Inside `<tool_call>`, each tool invocation is a JSON object with two keys: `name` (the tool name) and `parameters` (a JSON object mapping parameter names to parameter values). Multiple tool calls are written as multiple JSON objects (one per line) inside the same `<tool_call>` block. The following is a minimal example:

```
<tool_call>
{"name": "search_flights",
 "parameters": {"from": "SFO", "to": "JFK",
                "date": "2025-01-01"}}
{"name": "book_flight",
 "parameters": {"flight_id": "AA123", "seat": "12A"}}
</tool_call>
```

Parameter values are strings, numbers, booleans, lists, or nested JSON objects. During evaluation, tool names and parameter names/values are compared after parsing the JSON objects; outputs that violate the required field order or cannot be parsed receive zero format reward (Sec. J).

## J. Tool-Calling Rewards

We use two reward components for tool calling: a binary format reward and a graded correctness reward, following the GDPO evaluation protocol on BFCL-v3 (Liu et al., 2026; Patil et al., 2025).

**Format reward.** The format reward $R_{\text{format}} \in \{0, 1\}$ checks whether the output satisfies the required structure and contains all necessary fields in the correct order:

$$R_{\text{format}} = \begin{cases} 1, & \text{if all required fields appear and are in the correct order,} \\ 0, & \text{otherwise.} \end{cases} \tag{50}$$

**Correctness reward.** The correctness reward $R_{\text{correct}} \in [-3, 3]$ evaluates the predicted tool calls against the ground-truth calls. Let $\mathcal{G} = \{g_1, \ldots, g_{|\mathcal{G}|}\}$ denote the ground-truth tool calls and $\mathcal{P} = \{p_1, \ldots, p_{|\mathcal{P}|}\}$ denote the predicted tool calls parsed from `<tool_call>`. We score correctness by matching tool names, argument names, and argument values:

- **Tool name matching:** $s_{\text{name}} = \frac{|\text{names}(\mathcal{G}) \cap \text{names}(\mathcal{P})|}{|\text{names}(\mathcal{G}) \cup \text{names}(\mathcal{P})|} \in [0, 1]$.

- **Argument-name matching:** for a matched pair $(g, p)$, let $\text{keys}(g)$ and $\text{keys}(p)$ be their argument-name sets and score $\frac{|\text{keys}(g) \cap \text{keys}(p)|}{|\text{keys}(g) \cup \text{keys}(p)|}$; we sum this term over matched calls.

- **Argument-value matching:** for each matched pair $(g, p)$ and each key $k \in \text{keys}(g)$, add $\mathbf{1}[p[k] = g[k]]$; we sum this term over matched calls.

We then find the optimal matching between $\mathcal{G}$ and $\mathcal{P}$ that maximizes the total match score. Let $s_{\text{max}}$ denote the total match score under the optimal matching and let $s_{\text{max,possible}}$ denote the maximum achievable match score given $\mathcal{G}$. The final correctness reward is scaled to $[-3, 3]$ by

$$R_{\text{correct}} = 6 \cdot \frac{s_{\text{max}}}{s_{\text{max,possible}}} - 3. \tag{51}$$

**Length reward construction.** Following the GDPO/ToolRL setup (Liu et al., 2026; Qian et al., 2025), we add an explicit length term defined on the reasoning span inside `<think>`.[1] Let $L(x)$ be the `<think>`-word count of completion $x$. Reward-based baselines use the same settled/dynamic length reward:

$$R_{\text{len}}(x) = \min\{1, \ L(x)/L_{\text{max}}\} \in [0, 1], \tag{52}$$

followed by linear scaling to $[0, 1]$ (we set the max/min to 1/0 in our runs). Here $L_{\text{max}}$ is either a fixed target or a scheduled target:

$$L_{\text{max}} = \begin{cases} 200, & \text{short setting,} \\ 512, & \text{long setting (settled),} \\ 384 + (640 - 384) \cdot \frac{\text{step}}{105}, & \text{long setting (scheduled).} \end{cases} \tag{53}$$

In contrast, CoPro uses length as a feasibility constraint in the short setting by converting it into a nonnegative violation $g_{\text{len}}(x) = \max\{0, \ L(x) - 200\}$ and using the corresponding negative score $-g_{\text{len}}(x)$ in the update rather than optimizing a scalar "be longer" reward.

## K. Tool-Calling Training Details

Our tool-calling training follows the same VERL configuration used by GDPO/ToolRL (Liu et al., 2026; Qian et al., 2025) (Table 6). In our implementation, CoPro corresponds to setting `algorithm.adv_estimator=copro` in VERL. We train Qwen-2.5-Instruct models (1.5B and 3B) with group sampling (four rollouts per prompt) for 100 optimization steps (102 steps in our logs) and cap the maximum generated response length at 512 tokens. Evaluation is conducted on BFCL-v3 (Patil et al., 2025) and we report mean tool-calling accuracy and mean format correctness over 10 runs. We evaluate two length settings: a long setting with target length $l=512$ and a short setting with target length $l=200$; for CoPro, we enforce the short setting by directly constraining the output length rather than optimizing a length reward.

---

[1]In our implementation, length is measured by whitespace-separated *word count* in the extracted `<think>` string; missing `<think>` is treated as violating the requirement.

*Table 6.* Tool-calling training configuration (aligned with GDPO/ToolRL).

| Parameter | Value |
|---|---|
| trainer.total_epochs | 15 |
| data.train_batch_size | 512 |
| data.val_batch_size | 128 |
| data.max_prompt_length | 2048 |
| data.max_response_length (cap) | 512 |
| actor_rollout_ref.rollout.n (rollouts per prompt) | 4 |
| actor_rollout_ref.rollout.temperature / top_p | 1.0 / 1.0 |
| actor_rollout_ref.rollout.gpu_memory_utilization | 0.6 |
| actor_rollout_ref.actor.optim.lr | $1 \times 10^{-6}$ |
| actor_rollout_ref.actor.ppo_mini_batch_size | 128 |
| algorithm.kl_ctrl.kl_coef | 0.001 |

*Table 7.* AnalogGym sensitivity ablations on PFC. Each row varies one knob while keeping others fixed; **bold** marks the standard hyperparameter setting used in our main experiments.

| Knob | Setting | HV ↑ | VR ↓ | GCI ↓ |
|---|---|---|---|---|
| Group size $G$ | 4 / **8** / 16 | 0.590 / **0.683** / 0.686 | 0.110 / **0.058** / 0.060 | 0.045 / **0.021** / 0.022 |
| Distinctiveness ratio $\rho$ | 0 / 0.3 / **0.7** / 0.9 | 0.690 / 0.686 / **0.683** / 0.670 | 0.070 / 0.062 / **0.058** / 0.056 | 0.040 / 0.026 / **0.021** / 0.017 |
| Feasible anchor strength $\varepsilon_{\mathrm{feas}}$ | small / **mid** / large | 0.689 / **0.683** / 0.670 | 0.082 / **0.058** / 0.040 | 0.024 / **0.021** / 0.019 |
| E-step KL temperature $\tau_{\mathrm{KL}}$ | small / **mid** / large | 0.689 / **0.683** / 0.673 | 0.060 / **0.058** / 0.057 | 0.026 / **0.021** / 0.017 |
| M-step trust weight $\beta$ | small / **mid** / large | 0.688 / **0.683** / 0.670 | 0.070 / **0.058** / 0.045 | 0.023 / **0.021** / 0.020 |
| Base distribution $u$ | **uniform** / $\propto \pi_{\theta_{\mathrm{old}}}(x_i \mid c)$ | **0.683** / 0.676 | **0.058** / 0.054 | **0.021** / 0.020 |
| Dual update smoothing | none / **EMA** | 0.683 / **0.684** | 0.066 / **0.057** | 0.023 / **0.021** |

**GCI computation.** For the tool-calling curves, we compute the gradient cancellation index on the same trainable actor parameters used by the optimizer. For each logged group, we form the per-sample score-function gradients before the optimizer step, weight them by the corresponding signed or projected group weights, and then apply Eq. (9). This uses the actual training gradients rather than a low-dimensional proxy.

**Hypervolume for tool calling.** To summarize joint progress on correctness and format during training (Fig. 5a), we compute a 2D hypervolume over the reward vector $(R_{\mathrm{format}}, R_{\mathrm{correct}})$. Since $R_{\mathrm{format}} \in \{0, 1\}$ and $R_{\mathrm{correct}} \in [-3, 3]$, we rescale correctness to $[0, 1]$ by

$$\widetilde{R}_{\mathrm{correct}} := \frac{R_{\mathrm{correct}} + 3}{6} \in [0, 1]. \tag{54}$$

At each training step, we collect all rollout reward pairs $\{(R_{\mathrm{format}}(x), \widetilde{R}_{\mathrm{correct}}(x))\}$ from the logged rollouts (across prompts in the batch) and compute the hypervolume of the dominated region with respect to the reference point $r = (0, 0)$:

$$\mathrm{HV} := \mathrm{Area}\Big( \bigcup_{p \in \mathcal{P}} [0, p_1] \times [0, p_2] \Big), \qquad \mathcal{P} = \{(R_{\mathrm{format}}, \widetilde{R}_{\mathrm{correct}})\}. \tag{55}$$

Higher HV indicates better joint satisfaction of format and correctness without requiring an explicit scalarization.

## L. Additional Ablations (AnalogGym)

This section reports additional AnalogGym ablations that isolate the role of CoPro components. Unless stated otherwise, all ablations follow the same 4000-evaluation budget and report mean±std over seeds.

**Sensitivity ablations.** Table 7 varies key hyperparameters and modeling choices that control conservativeness and stability, including group size $G$, the distinctiveness ratio $\rho$ (Eq. (13)), feasible anchor strength $\varepsilon_{\mathrm{feas}}$ (Appendix A), and the E/M-step regularization weights $(\tau_{\mathrm{KL}}, \beta)$.

**Mechanistic reading of Table 7.** *Group size $G$* controls the number of within-group pairwise comparisons that define the comparative statistics and the E-step moments; smaller groups yield a weaker and noisier ordering signal, which propagates to a less stable $q^*$ and a higher fraction of infeasible updates (lower HV, higher VR, higher GCI). *Distinctiveness ratio $\rho$* sets

*Table 8.* Residual-design ablation on PFC (AnalogGym, 4000 SPICE evaluations). The piecewise-log residual is the default used by CoPro.

| Residual design $g(x)$ | HV $\uparrow$ | VR $\downarrow$ | GCI $\downarrow$ |
|---|---|---|---|
| piecewise-log (ours) | **0.683** | **0.058** | **0.021** |
| normalized linear map | 0.675 | 0.062 | 0.024 |
| clipped normalized linear map | 0.665 | 0.068 | 0.027 |
| unnormalized linear map | 0.643 | 0.078 | 0.035 |

*Table 9.* Residual-design diagnostics on BFCL-v3 with Qwen-2.5-Instruct 1.5B (short setting).

| Residual design $g(x)$ | Acc. $\uparrow$ | Format $\uparrow$ | $q(\mathcal{F})\uparrow$ | ESS $\uparrow$ |
|---|---|---|---|---|
| piecewise-log (ours) | **30.1%** | **83.1%** | **0.80** | **3.20** |
| normalized linear map | 29.7% | 81.8% | 0.76 | 3.00 |
| clipped normalized linear map | 29.2% | 80.7% | 0.73 | 2.85 |
| unnormalized linear map | 27.2% | 76.4% | 0.61 | 2.42 |

the adaptive threshold $\tau_c = \rho \max_i h_i$ and therefore directly determines how aggressively the E-step suppresses near-ties; increasing $\rho$ reduces near-tie-driven concentration and lowers GCI, while overly large $\rho$ limits weight on high-scoring candidates and reduces HV under a fixed budget. *Feasible anchor strength* $\varepsilon_{\mathrm{feas}}$ controls the margin of the feasibility residual used in $g$; larger $\varepsilon_{\mathrm{feas}}$ increases the anchoring effect and reduces VR by biasing $q^*$ toward near-feasible anchors, while overly large anchoring reduces objective-side progress and lowers HV. *E-step temperature* $\tau_{\mathrm{KL}}$ sets the sharpness of the exponential reweighting: smaller $\tau_{\mathrm{KL}}$ produces highly concentrated $q^*$ that amplifies noise in comparisons, while larger $\tau_{\mathrm{KL}}$ produces smoother weights that reduce cancellation and stabilize VR but slow objective progress. *M-step trust weight* $\beta$ controls how tightly the policy is constrained to the previous iterate; larger $\beta$ reduces constraint-violation spikes by limiting distribution shift, while smaller $\beta$ increases step size and yields more volatile VR and GCI. *Base distribution* $u$ and *dual smoothing* act on the same failure mode: high-variance per-group reweighting. A uniform $u$ preserves exploration, while likelihood-shaped $u$ concentrates on the current policy mode; EMA smoothing stabilizes the dual dynamics and reduces oscillation in VR/GCI driven by noisy group feedback.

**Residual-design ablations.** We further isolate the infeasibility-side residual mapping $g(x)$ while keeping the rest of CoPro fixed. Tables 8 and 9 show that the exact functional form is not unique, but scale-aware compression improves the feasibility-quality trade-off and stabilizes the E-step projection.

## M. Diagnostic Ablations (Tool Calling)

For tool calling, we report *diagnostic* ablations that correspond to CoPro's design motivations. In addition to the standard metrics (accuracy, format compliance, and length), we report per-group quantities derived from the solved reweighting distribution $q^*$:

- **Anchor mass:** $q(\mathcal{F}) = \sum_{i:g_i \le 0} q_i$.

- **Effective sample size (ESS):** $\mathrm{ESS}(q) = 1/\sum_i q_i^2$.

- **Weight entropy:** $H(q) = -\sum_i q_i \log q_i$.

Table 10 summarizes a compact diagnostic suite suitable for BFCL-style tool calling.

**Mechanistic reading of Table 10.** CoPro treats length control as a feasibility constraint in the short setting; the feasibility moment therefore anchors $q^*$ on length-feasible samples and prevents the update from being dominated by longer completions that accumulate more format failure opportunities. Removing this moment shifts optimization pressure toward length as a reward, increasing format violations and reducing $q(\mathcal{F})$. The distinctiveness moment targets near-ties in multi-reward

*Table 10.* Tool-calling diagnostic ablations on BFCL-v3 with Qwen-2.5-Instruct 1.5B (short setting). In addition to accuracy/format, we log anchor mass $q(\mathcal{F})$ and effective sample size (ESS) to diagnose feasibility anchoring and near-tie concentration.

| Variant | Acc. ↑ | Format ↑ | $q(\mathcal{F})$ ↑ | ESS ↑ |
|---|---|---|---|---|
| CoPro (full) | 30.1% | 83.1% | 0.80 | 3.20 |
| w/o feasibility moment (optimize length as reward) | 26.8% | 72.6% | 0.18 | 2.40 |
| w/o distinctiveness moment | 28.8% | 76.4% | 0.78 | 1.50 |
| w/o nonnegative projection (signed weights) | 26.2% | 68.9% | 0.76 | 1.20 |

*Table 11.* Tool-calling results when length is enforced as a hard constraint for all methods (instead of being optimized as a reward term). All methods satisfy the length budgets, but most reward-based baselines incur a 1–3% performance drop, while LCPO remains stable.

| **Metric** | | **CoPro** | **DAPO** | **GDPO** | **LCPO** | **GRPO** |
|---|---|---|---|---|---|---|
| **Qwen-2.5** | Avg Acc ↑ | **31.7%** | 28.2% | **30.0%** | 29.5% | 26.5% |
| **1.5B** | Fmt Acc ↑ | **83.1%** | 79.0% | **79.6%** | 79.2% | 76.0% |
| **(Short)** | Avg Length | 124.6 | 129.8 | 130.5 | 136.9 | 131.2 |
| **Qwen-2.5** | Avg Acc ↑ | **33.9%** | 29.4% | 30.6% | **30.9%** | 28.2% |
| **1.5B** | Fmt Acc ↑ | **83.5%** | 80.0% | **80.1%** | 79.5% | 77.7% |
| **(Long)** | Avg Length | 468.2 | 485.0 | 491.0 | 441.8 | 496.0 |
| **Qwen-2.5** | Avg Acc ↑ | **40.8%** | 37.0% | **38.7%** | 38.6% | 35.2% |
| **3B** | Fmt Acc ↑ | **83.7%** | 80.8% | **81.2%** | 81.0% | 79.1% |
| **(Short)** | Avg Length | 126.3 | 130.0 | 131.0 | 139.1 | 131.5 |
| **Qwen-2.5** | Avg Acc ↑ | **42.6%** | 38.2% | **39.8%** | 39.7% | 36.3% |
| **3B** | Fmt Acc ↑ | **84.9%** | 82.0% | **82.2%** | 81.9% | 80.2% |
| **(Long)** | Avg Length | 472.5 | 489.0 | 494.0 | 448.7 | 500.0 |

comparisons (correctness vs. format under sparse binary feedback). Removing it concentrates weight on groups with ambiguous comparisons, lowering ESS and increasing update variance on shared tool schemas, which reduces format compliance. Signed, normalized advantages couple correctness, format, and length into a single scalar with sign flips across prompts; these sign flips induce cancellation in shared parameters, which degrades both accuracy and format and yields more concentrated weights (lower ESS).

**Length as a hard constraint for baselines.** We next test whether the observed length behavior is specific to CoPro's feasibility formulation or simply an artifact of how length is handled during training. Concretely, we re-run the baselines by treating length as a hard feasibility condition (analogous to $g(x) \leq 0$): a completion is deemed feasible if its length satisfies the setting-specific budget, and infeasible samples do not contribute to group updates. Table 11 shows that all methods can satisfy the length budgets under this constraint-only treatment, but DAPO/GDPO/GRPO incur a consistent 1–3% degradation in accuracy and/or format compliance, while LCPO remains essentially unchanged.

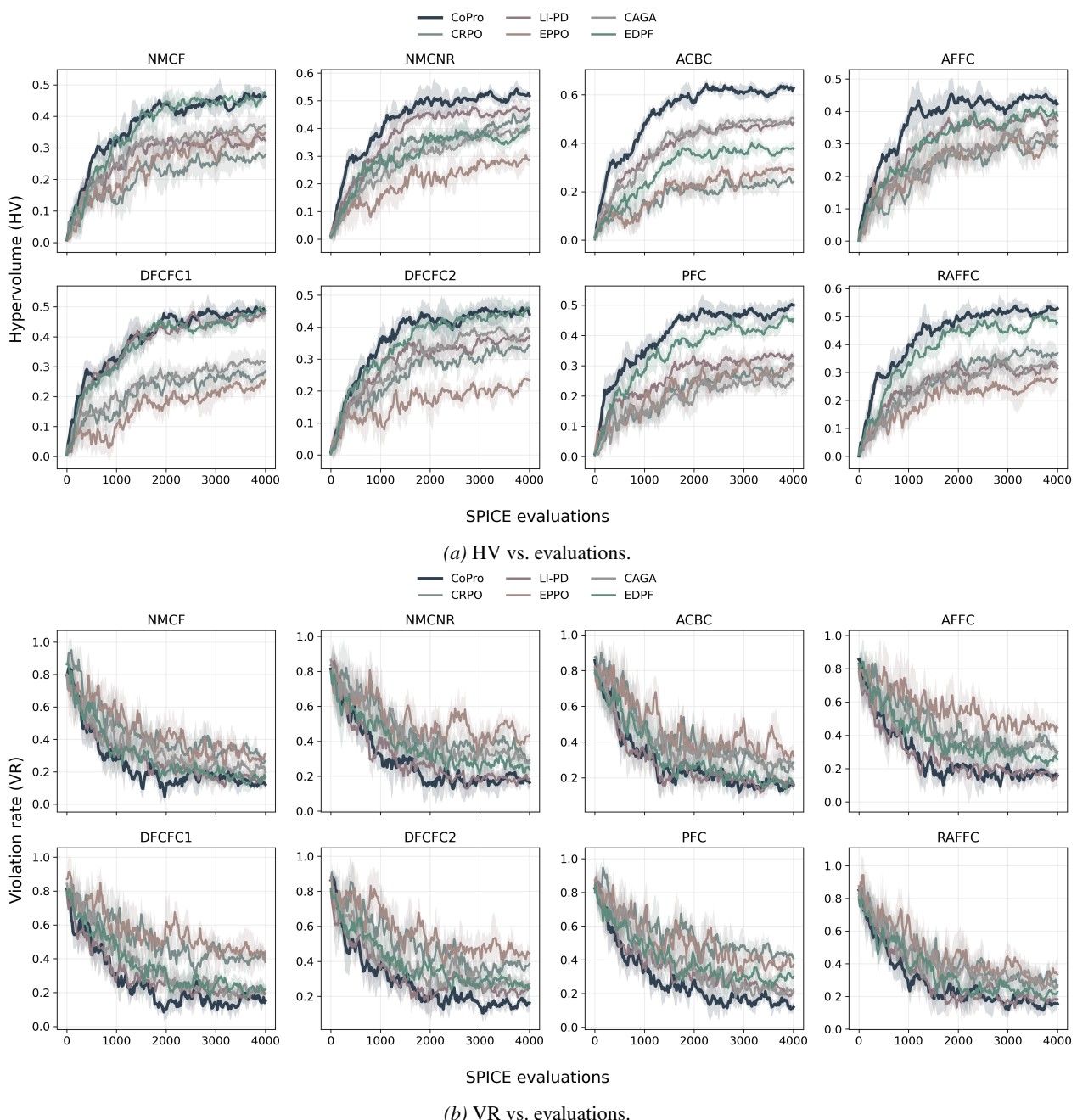

*(a)* HV vs. evaluations.

*(b)* VR vs. evaluations.

*Figure 8.* Additional constrained MORL comparisons on AnalogGym. We report HV and VR across eight circuits (NMCF, NMCNR, ACBC, AFFC, DFCFC1, DFCFC2, PFC, RAFFC) for CoPro and constrained baselines (CRPO, LI-PD, EPPO, CAGA, EDPF).

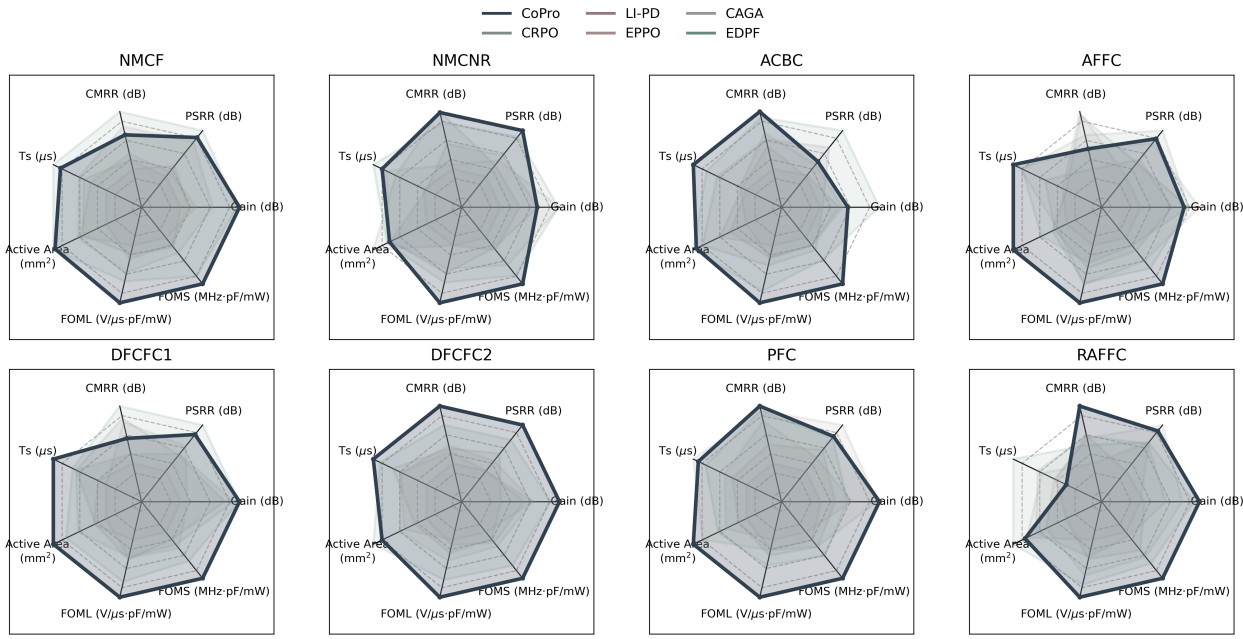

*Figure 9.* Radar comparison on eight AnalogGym circuits. Each subplot corresponds to one circuit and aggregates multiple specifications/objectives into a normalized radar profile (higher is better after direction-aware normalization).

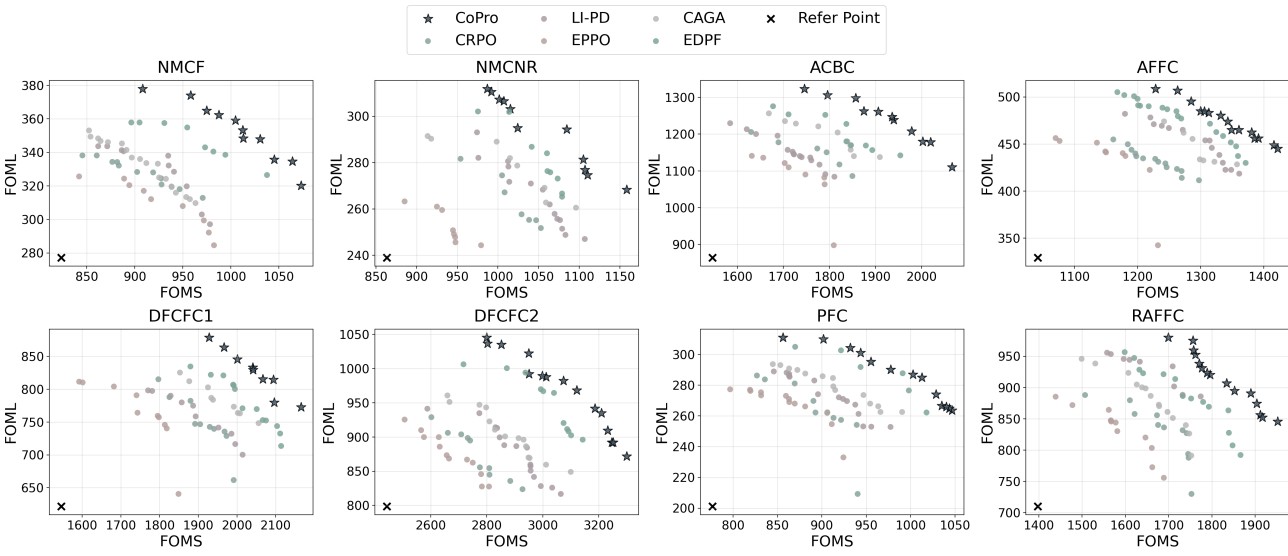

*Figure 10.* Feasible Pareto fronts in $(\mathrm{FOM}_S, \mathrm{FOM}_L)$ space across eight AnalogGym circuits, with FOMS on the x-axis and FOML on the y-axis. Colors match the radar/curve plots.

