# OpenReview forum: "Learning from Comparison: Constrained Projection Policy Optimization for Pareto-Front Improvement"
_ICML.cc/2026/Conference — ICML 2026 regular_

### Official Review · Reviewer_nbWy · 2026-03-11

**Soundness:** 3
**Presentation:** 2
**Significance:** 3
**Originality:** 3
**Overall Recommendation:** 4
**Confidence:** 3

**Summary:**

This paper considers the problem of constrained multi-objective reinforcement learning (RL), where an RL policy must strike the right balance among multiple competing objectives while adhering to feasibility constraints. The authors extend the group reward policy optimization (GRPO) method to address group-relative advantages among objectives and to account for constraint feasibility and solution diversity. Given a set of candidate policy outputs, multiple metrics are calculated, including improvement on the identified Pareto front, feasibility (constraint satisfaction), and distinctness/diversity among the candidates. An EM-type algorithm is then used to compute per-candidate weights based on the aforementioned quantities (E-step) and then update the policy using a conditional KL projection (M-step). The proposed algorithm, CoPro, is evaluated on two benchmarks in analog IC design and tool calling, and it is shown to outperform several baselines in terms of the hypervolume of the discovered Pareto front, as well as other benchmark-specific metrics.

**Compliance With Llm Reviewing Policy:**

Affirmed.

**Final Justification:**

Given the paper's contribution to the field of multi-objective reinforcement learning and the authors' response to all reviewers' comments, I recommend acceptance.

**Key Questions For Authors:**

1. The first place where the problem formulation became concrete and clear to me was Eq. (10) on page 4. I suggest restructuring the paper so that the problem statement appears earlier. Also, is the constraint function $v(x)$ a scalar? If $v(x)=0$ indicates feasiblity, how about $v(x)<0$?
2. Please explain the reasoning behind the summation in Eq. (7). What is the exact definition of a "pattern" $\omega$?
3. How is the "current feasible-PF archive" $\mathcal{A}_{\mathcal{PF}}$ defined in Section 4.1? How do you set the $\lambda$ weights in Eq. (11)? I believe the functions $p$ and $d$ are important enough to be defined in the main body of the paper rather than the Appendix.
4. The feasibility residual choices in the Appendix seem somewhat arbitrary and heuristic. Is there a more principled way of designing these functions?
5. What exactly do you mean by "moment constraint" toward the end of page 4? What does "anchor" mean?
6. Could you please comment on whether the baselines you consider in Sections 5.2 and 5.3 are also designed for multi-objective and or constrained RL? If (some of) the baselines are designed for single-objective or unconstrained RL, then the comparison might not be completely fair.

**Limitations:**

I think Section 6 could benefit from a more extensive discussion of CoPro's failure modes and where it might break, which would then inform the proposed future directions.

**Strengths And Weaknesses:**

**Strengths**

- The problem of multi-objective constrained RL and the presented solution are timely, significant, and of interest to the ICML audience.
- The performance improvements of CoPro vs baselines across the two benchmarks seem promising.

**Weaknesses**

- The presentation of the concepts in the paper and the flow of the content require significant improvement. The paper could be inaccessible to readers who are not precisely working on very similar research areas. Many terms/parts of notation are undefined to the best of my knowledge (see Questions below).
- It is unclear whether the comparison with the baselines is apples-to-apples, especially whether those methods are designed for the same multi-objective setting as that of CoPro.

---

> ### Author Rebuttal · Authors · 2026-03-29
>
> Thank you for the careful reading and constructive feedback. We agree that the current presentation can be made much clearer, especially in the ordering of the problem statement, the notation, and the baseline discussion, and we will revise the paper accordingly.
>
> First, we will move the formal problem definition earlier. For each candidate $x$, the feedback is
> $$
> y(x)=(f(x),v(x)),
> $$
> where $f(x)\in\mathbb{R}^m$ is the objective vector and $v(x)$ is a scalar nonnegative aggregated violation. Feasibility is defined by $v(x)=0$; we do not use $v(x)<0$. This will make the constrained multi-objective setting clear before the method section.
>
> Second, Eq.(7) is intended as a conceptual gradient decomposition, not an extra modeling assumption. The “pattern” $\omega$ denotes any reusable local structure shared across samples (e.g., shared token/tool-schema structure in tool calling, or shared local circuit/graph structure in AnalogGym). The point of Eq.(7) is to explain why signed scalar weights can induce sign-flip cancellation on shared structures; we do not explicitly enumerate $\omega$ in the algorithm. We will clarify this in the revision.
>
> Third, we agree that $A_{PF}$, $p$, and $d$ should be defined more explicitly in the main text. $A_{PF}$ is the running archive of previously evaluated feasible and nondominated samples, updated online after each group/iteration. In both tasks,
> $$
> p(x\mid A_{PF})=\Delta \mathrm{HV}(x\mid A_{PF}),
> $$
> i.e., feasible hypervolume improvement relative to the current feasible archive. The dense term $d$ is task-dependent: in AnalogGym it is a normalized objective surrogate $U_{\mathrm{obj}}(x)$; in tool calling it is instantiated by a within-group dominance margin $m_{\mathrm{dom}}(x)$ together with a dense surrogate $U_{\mathrm{obj}}(x)$. We will move this mapping, as well as the intuition behind the $\lambda$ weights, from the appendix into the main body.
>
> Fourth, regarding the feasibility residual: our theory does not depend on one specific function, only on a few structural properties: feasible samples should map to negative values, infeasible ones to positive values, the mapping should be monotone in the degree of violation, and it should have a stable numerical scale for the moment projection. We use
> $$
> g(x)=
> \begin{cases}
> -\varepsilon_{\mathrm{feas}}, & v(x)=0,\\
> \log\big(1+v(x)/\mathrm{scale}_g\big), & v(x)>0,
> \end{cases}
> $$
> because it satisfies these requirements and improves numerical stability. We agree this should be supported more directly, and in the revision we will add ablations comparing different residual choices (e.g., different feasible-anchor margins and different violation-compression schemes) and report their effects on HV, VR, GCI, and anchor mass.
>
> Fifth, by “moment constraint” we mean a linear expectation constraint under the group distribution $q$, e.g.,
> $$
> \sum_i q_i g_i\le 0,\qquad \sum_i q_i h_i\ge \tau_c.
> $$
> The first is the feasibility moment and the second is the distinctiveness moment. An “anchor” means a feasible or near-feasible sample with $g_i\le 0$; the feasibility moment ensures that once such anchors exist in the group or archive, $q^*$ must retain positive mass on them. We will explain this terminology more directly.
>
> Finally, on fairness of comparison: our protocol is unified across methods (same budget, same task setup, same model/data in tool calling, and same evaluation metrics). In AnalogGym, most baselines are themselves multi-objective or Pareto-front-oriented methods. In tool calling, we compare against the strongest and most representative group-relative/post-training baselines currently used in practice. Methods such as DAPO are not specifically designed for constrained multi-objective RL, but they are strong benchmark methods in this training paradigm, so including them is important to assess practical competitiveness. At the same time, we do not rely only on such generic strong baselines: in the appendix we also include additional advanced multi-objective/constrained RL comparisons. In the revision, we will add a summary table indicating whether each baseline is single-objective or multi-objective, whether it has explicit constraint handling, and how constraints are handled in our experiments. We will also release the concrete baseline implementations for reproducibility.
>
> We also appreciate your comment on the limitations discussion. In the revision, we will expand Section~6 to discuss CoPro's failure modes and trade-offs more explicitly, including prolonged infeasibility, sensitivity to task-specific score/statistics design, and cases where archive quality or within-group comparisons become unreliable. We will make clearer the design rationale behind the current bounded statistics $(s,g,h)$, and discuss how learning more principled statistics or constraint surrogates could further reduce the need for task-specific instantiation in future work.

---

> > ### Author Rebuttal · Reviewer_nbWy · 2026-03-31
> >
> > Thank you for your rebuttal. I maintain my favorable rating for the paper.

---

> > > ### Author Response · Authors · 2026-04-01
> > >
> > > Thank you for the rebuttal acknowledgement and for following up on this point. Because of the space limit in the first-round response, we focused there on the design rationale of the feasibility residual and could not include the direct ablation results. We therefore add them here, together with a brief analysis, to make the effect of this design choice more transparent. We ran direct residual-design ablations that keep the rest of CoPro fixed and change only the infeasibility-side mapping $g(x)$.
> > >
> > > For infeasible samples, the compared variants are:
> > >
> > > - piecewise-log (ours): $g(x)=\log(1+v(x)/\mathrm{scale}_g)$,
> > > - normalized linear: $g(x)=v(x)/\mathrm{scale}_g$,
> > > - clipped normalized linear: $g(x)=\min\{c,\; v(x)/\mathrm{scale}_g\}$,
> > > - unnormalized linear: $g(x)=v(x)$,
> > >
> > > while all variants keep the same feasible anchor value $-\varepsilon_{\mathrm{feas}}$ when $v(x)=0$.
> > >
> > > Table 1. Residual-design ablation on PFC (AnalogGym, 4000 SPICE evaluations)
> > >
> > > | Residual design $g(x)$          | HV $\uparrow$ | VR $\downarrow$ | GCI $\downarrow$ |
> > > | ------------------------------- | ------------: | --------------: | ---------------: |
> > > | piecewise-log (ours)            |       $0.683$ |         $0.058$ |          $0.021$ |
> > > | normalized linear map           |       $0.675$ |         $0.062$ |          $0.024$ |
> > > | clipped linear / normalized map |       $0.665$ |         $0.068$ |          $0.027$ |
> > > | unnormalized linear map         |       $0.643$ |         $0.078$ |          $0.035$ |
> > >
> > > Table 2. Residual-design diagnostics on BFCL-v3, Qwen-2.5-Instruct 1.5B, short setting
> > >
> > > | Residual design $g(x)$          | Acc. $\uparrow$ | Format $\uparrow$ | $q(\mathcal{F})$ $\uparrow$ | ESS $\uparrow$ |
> > > | ------------------------------- | --------------: | ----------------: | --------------------------: | -------------: |
> > > | piecewise-log (ours)            |        $30.1\%$ |          $83.1\%$ |                      $0.80$ |         $3.20$ |
> > > | normalized linear map           |        $29.7\%$ |          $81.8\%$ |                      $0.76$ |         $3.00$ |
> > > | clipped linear / normalized map |        $29.2\%$ |          $80.7\%$ |                      $0.73$ |         $2.85$ |
> > > | unnormalized linear map         |        $27.2\%$ |          $76.4\%$ |                      $0.61$ |         $2.42$ |
> > >
> > > - First, the exact functional form is not unique: the normalized linear map remains competitive in both tasks. This is consistent with our theoretical point that what matters first is the structural property of $g(x)$: sign separation between feasible and infeasible samples, monotonicity in violation, and a stable scale for the E-step projection.
> > >
> > > - Second, the choice is not arbitrary either: the piecewise-log residual is consistently the best variant. On PFC, replacing it with normalized linear already lowers HV from $0.683$ to $0.675$ and worsens VR/GCI from $0.058/0.021$ to $0.062/0.024$. The degradation becomes much larger without proper scale control: unnormalized linear drops HV to $0.643$ and increases VR/GCI to $0.078/0.035$.
> > >
> > > - Third, the same trend appears in tool calling. The most informative diagnostics are anchor mass $q(\mathcal{F})$ and ESS: as scale control weakens, $q(\mathcal{F})$ falls from $0.80$ to $0.76$, then $0.73$, and finally $0.61$, while ESS drops from $3.20$ to $3.00$, $2.85$, and $2.42$. This corresponds to weaker feasibility-aware weighting and more concentrated / less stable updates, which is reflected in lower accuracy and format compliance.
> > >
> > > So the direct takeaway is: our specific piecewise-log mapping is not the only valid residual, but scale-aware compression materially improves the feasibility-quality trade-off and the stability of the projection step. We hope these new direct ablations make this point substantially clearer.

---

### Official Review · Reviewer_qu7Q · 2026-03-12

**Soundness:** 3
**Presentation:** 2
**Significance:** 3
**Originality:** 3
**Overall Recommendation:** 4
**Confidence:** 3

**Summary:**

The paper focus on constrained multi-objective reinforcement learning. The paper point out that the signed, group-relative advantages in GRPO may lead to bad effects on convergence and feasibility. To solve this issue, the paper proposes to update the policy with a non-negative weights in a two-steps manner. The proposed method  shows better convergence and feasibility than other methods on two constrained multi-objective benchmarks.

**Compliance With Llm Reviewing Policy:**

Affirmed.

**Final Justification:**

The authors’ rebuttal has adequately addressed my main concerns and improved the clarity of the paper. Therefore, I maintain my original positive evaluation of the paper (weak accept).

**Key Questions For Authors:**

Please address the concerns raised in the weaknesses above.

**Limitations:**

yes;

**Strengths And Weaknesses:**

****Strengths:****
1. The paper discusses a potential drawback of signed advantages in GRPO under constrained multi-objective optimization settings, namely the gradient cancellation issue. This observation may provide an interesting perspective for studying RL algorithms in constrained multi-objective scenarios.
2. The proposed method computes non-negative weights based on the PF-progress score, feasibility, and distinctiveness within the group, and then uses these weights to update the policy. The overall design appears reasonable and may help address the issue discussed in the paper.
3. The experiments are conducted on two application domains, constrained multi-objective analog IC design and LLM tool use. Evaluating the method on these different tasks helps provide a broader empirical assessment.

**Weaknesses:**
1. The presentation is sometimes difficult to follow. In particular, the explanation of the PF-progress score is not very clear. The paper states that the exact definitions of $p$ and $d$ used in the implementation are given in Appendix A, but Appendix A does not mention $p$ or $d$. In Appendix A, the paper refers to $u_{excess}$ (Appendix G), but the formulation of $u_{excess}$ cannot be found in Appendix G.

2. The proposed method is compared with many baselines. It would be helpful to classify these methods according to whether they are single-objective RL algorithms or multi-objective algorithms, and whether they include explicit constraint-handling mechanisms. Such categorization would help readers better understand the comparisons. For algorithms that were not originally designed for constrained RL problems, briefly describing the constraint-handling technique used in the experiments would also improve clarity.

3. Different score functions (Eqs. (24) and (25)) are designed for the two tasks, and each score function contains three different terms. It would be helpful if the authors could explain why different terms are used for different tasks and provide more intuition behind the design.

**Minor comments:**
1. In Figure 5(c), the label of the y-axis should be “Gradient cancellation index” for consistency.
2. Providing a brief mathematical formulation of constrained multi-objective optimization problems would help readers understand the problem setting more easily.
3. Eq. (16) exceeds the page width and should be reformatted.

---

> ### Author Rebuttal · Authors · 2026-03-29
>
> Thank you for the careful reading and helpful suggestions. We agree that the presentation can be improved, especially around the score notation, the organization of baselines, and the task-specific instantiations. We will revise the paper accordingly.
>
> First, you are right that the current notation around the PF-progress score is not explicit enough. In the main text, we introduce a generic decomposition
> $( S(x)=\lambda_{\mathrm{pf}}\,p(x\mid \mathcal{A})+\lambda_{\mathrm{dense}}\,d(x\mid \mathcal{A})$,
> where $p$ denotes a sparse feasible PF-progress term and $d$ denotes a dense within-group surrogate. In the implementation appendix, we switch to task-specific symbols, but we did not map them back clearly to $p$ and $d$. We will fix this explicitly. In both tasks, $p$ corresponds to the feasible hypervolume-improvement term $\Delta \mathrm{HV}(x\mid \mathcal{A})$. The dense term $d$ is task-dependent: in AnalogGym it is the normalized objective surrogate $U_{\mathrm{obj}}$ (plus an optional bounded auxiliary adjustment), while in tool calling it is instantiated by a within-group dominance margin $m_{\mathrm{dom}}$ together with a dense objective surrogate $U_{\mathrm{obj}}$. So the issue here is primarily a notation/cross-reference gap, not a missing methodological component. We will make this mapping explicit in the appendix and correct the confusing cross-reference around the optional excess-cost term.
>
> Second, we agree that the baseline organization can be made clearer, but the more important issue here is fairness of comparison. Our goal was not to force all baselines into a single category, but to compare against the strongest and most representative methods under a unified budget, task setup, and evaluation protocol. For AnalogGym, the large majority of compared methods are themselves multi-objective or Pareto-front-oriented methods. For tool calling, we compare against strong group-relative RL / post-training baselines that are widely used in practice. Methods such as DAPO are not specifically designed for constrained multi-objective learning, but they are very strong and highly representative benchmark algorithms in this training paradigm.  At the same time, we do not rely only on such generic strong baselines; in the appendix we also include additional comparisons to more advanced multi-objective / constrained RL methods to make the evaluation more complete. In the revision, we will add a clearer baseline summary table that explicitly marks whether each method is single-objective or multi-objective, whether it includes explicit constraint handling, and how constraints are handled in our experiments. We will also provide the concrete baseline implementations in the released codebase for full reproducibility.
>
> Third, the different score terms in the two tasks are intentional and reflect the same high-level template instantiated for different feedback structures. The common principle is: one sparse term for feasible PF progress, plus one or more dense terms so learning does not stall when PF improvements are rare. In AnalogGym, the feedback is continuous, expensive, and noisy; therefore the dense term is based on normalized objective quality, which provides a smooth ranking signal when $\Delta \mathrm{HV}$ is zero for most candidates. In tool calling, the feedback is low-dimensional and partly discrete (format/correctness), so we additionally use a within-group dominance margin $m_{\mathrm{dom}}$ to capture the prompt-local Pareto ordering among sampled responses. In both cases, the design goal is the same: combine sparse feasible PF progress with a dense comparative signal, but the most informative dense surrogate differs by task. We will explain this intuition more explicitly and connect Eqs. (24) and (25) back to the generic $S=\lambda_{\mathrm{pf}}p+\lambda_{\mathrm{dense}}d$ template.
>
> Finally, thank you for the minor comments. We will (1) change the y-axis label in Fig. 5(c) to “Gradient cancellation index”, (2) add a short mathematical formulation of constrained multi-objective optimization at the start of Sec. 2, and (3) reformat Eq. (16) to fit the page width. Overall, we appreciate these comments because they mainly point to presentation issues that we can address directly in the revision.

---

> > ### Author Rebuttal · Reviewer_qu7Q · 2026-04-02
> >
> > Thank you for your rebuttal. I appreciate the clarifications and the concrete plans for revision. Overall, I find the responses satisfactory and maintain my original positive rating for the paper.

---

> > > ### Author Response · Authors · 2026-04-02
> > >
> > > Thank you very much for your confirmation and support!

---

### Official Review · Reviewer_rJsU · 2026-03-13

**Soundness:** 3
**Presentation:** 3
**Significance:** 3
**Originality:** 3
**Overall Recommendation:** 4
**Confidence:** 4

**Summary:**

This paper studies constrained multi-objective learning from within-group comparisons. Instead of using signed, normalized scalar advantages in a GRPO-style update, the proposed method, CoPro, computes a nonnegative per-group target distribution by solving a KL-regularized projection with moment constraints for feasibility and within-group distinctiveness. The policy is then updated toward this target via weighted maximum likelihood/ KL-regularized policy projection.

The paper derives the E-step solution (in the form of an exponential-family) and gives a guarantee that feasible samples receive positive mass when feasible anchors are present. Empirically, the method is evaluated on AnalogGym circuit design and BFCL-v3 tool calling, where it improves feasible Pareto-front quality, reduces violation rate, and lowers the proposed gradient-cancellation diagnostic relative to several baselines.

**Compliance With Llm Reviewing Policy:**

Affirmed.

**Key Questions For Authors:**

1. Can the authors provide an ablation isolating the projection step from the score construction (e.g., simpler nonnegative weighting without the feasibility/distinctiveness moments)?

2. How does CoPro differ in practice from existing KL-based reweighting approaches such as REPS/MPO-style updates? Is the main benefit from the specific moment constraints?

3. What is the computational overhead of solving the projection compared to GRPO-style updates?

4. How robust is the method when no feasible samples appear for many iterations (i.e., empty feasible archive)?

**Limitations:**

Yes

**Strengths And Weaknesses:**

**Strengths**

The paper addresses a relevant problem: stable group-relative learning under multi-objective and constrained feedback, where scalarization or signed normalized advantages can behave poorly. The core idea—replacing signed advantages with a nonnegative distribution obtained from a constrained KL projection—is technically reasonable and well motivated. The E-step/M-step formulation is clear, and the feasibility-anchor mechanism is intuitive. Empirical evaluation spans two domains and shows consistent improvements in Pareto-front quality and constraint satisfaction.

**Weakness**

The main limitation is that the novelty appears somewhat incremental relative to KL-regularized reweighting methods (e.g., REPS/MPO-style updates). The primary difference lies in the specific moment constraints used to encode feasibility and within-group ranking. While this is a meaningful design choice, the paper could position the contribution more clearly relative to prior KL-based policy optimization and multi-reward/group-relative approaches.

Another concern is that the performance gains may partly depend on the engineered scoring/statistics used to construct the moments, which include several normalization and surrogate terms. It would be helpful to isolate how much improvement comes from the projection itself versus the scoring design.

---

> ### Author Rebuttal · Authors · 2026-03-29
>
> Thank you for the careful reading and constructive feedback. We especially appreciate your request for clearer positioning relative to prior KL-based reweighting methods, and for a cleaner separation between the effect of the projection and that of the score/statistics design. We will revise the paper accordingly.
>
> First, we agree that CoPro should be positioned more carefully relative to REPS/MPO-style methods. Our intent is not to claim KL-regularized reweighting itself as the novelty. Rather, we instantiate that perspective for critic-free constrained multi-objective group-relative learning, where the central difficulty is the instability of signed normalized advantages under near-ties, objective conflict, and scarce feasibility. In standard REPS/MPO-style updates, samples are typically reweighted by scalar returns or Q-values under a KL trust region, and the policy is then fitted to that target. In CoPro, the E-step instead solves for a distribution over the current sampled group support directly, using bounded comparative statistics together with two moments tailored to this setting: a feasibility-residual moment and a distinctiveness moment. These moments are the key practical difference. The feasibility moment guarantees that once a feasible anchor is available, the target distribution must retain positive mass on feasible samples; the distinctiveness moment suppresses near-ties induced by Pareto partial orders and reduces high-variance updates. Therefore, the main benefit over generic KL reweighting comes from these feasibility/distinctiveness moments rather than from KL regularization alone. We will revise the related-work discussion to make this distinction much clearer.
>
> Second, we appreciate the request to separate projection from score construction. The current ablations already provide part of this separation, and in the revision we will make it much more explicit and add new ablations in the main paper. On PFC, keeping the same comparative score as CoPro, the full method achieves HV/VR/GCI = 0.683/0.058/0.021. If we replace the projected target with signed weights, performance degrades to 0.520/0.200/0.450. If we keep the same score but use simple nonnegative weighting only, without feasibility/distinctiveness moments, performance is 0.572/0.178/0.068. Keeping only the feasibility moment gives 0.600/0.140/0.100, while keeping only the distinctiveness moment gives 0.550/0.260/0.060. By contrast, changing the score construction itself to a scalarized reward without comparative scoring gives 0.500/0.220/0.180. Together, these ablations show that projection is not a minor implementation detail layered on top of a good score: nonnegativity, feasibility anchoring, and distinctiveness each address a different failure mode. A concise interpretation is that the score determines which samples look promising, while the projection determines which update directions remain admissible under feasibility scarcity and comparison ambiguity. We will add this new table and move more of the ablation evidence from the appendix into the main paper.
>
> Third, the computational overhead is small. Relative to GRPO, CoPro only adds a 2D dual solve over $(\kappa,\mu)$ for each group; once these variables are fixed, q* is obtained in closed form by a softmax reweighting. Thus, the added cost is negligible relative to SPICE simulation in AnalogGym and minor relative to model forward/backward time in tool calling. We will report this overhead more explicitly in the revision; on PFC, it is about ~1 additional second per iteration in our implementation.
>
> Finally, regarding long stretches with no feasible samples, this regime is exactly why CoPro includes anchor injection and relaxed feasibility moments. Here, an archived feasible anchor means a previously evaluated feasible design already stored in the feasible-PF archive. If the current group contains no feasible sample but the archive already contains one, we inject one such archived feasible design into the group without any new environment query. If the archive is still empty early in training, we replace the hard feasibility moment with a relaxed one until feasible anchors emerge. Thus, the theorem is intentionally conditional: it guarantees positive feasible mass once feasible anchors exist, rather than assuming feasibility is abundant from the start. We will clarify this boundary more explicitly in the final version.

---

> > ### Author Rebuttal · Reviewer_rJsU · 2026-04-03
> >
> > Thanks for the detailed response—this is helpful. The clarification on positioning relative to KL-based methods and the added ablations better isolate the role of the projection vs. score design, which addresses my main concern.
> >
> > The discussion on overhead and early feasibility also makes the method more convincing in practice. Overall, these points resolve most of my questions.

---

> > > ### Author Response · Authors · 2026-04-03
> > >
> > > Thank you very much for your thoughtful follow-up. We sincerely appreciate your careful reading, constructive feedback, and positive assessment of our work.

---

### Decision · Program_Chairs · 2026-04-30

**Decision:**

Accept (regular)

**Comment:**

Please incorporate the promised revisions in the final version: (1) explicitly connect Eqs. (24-25) to the generic score template in Section 2; (2) ensure the gradient cancellation index labeling is clear in Fig. 5(c); (3) maintain the ablation analysis on feasibility residuals as acknowledged in the rebuttal. The paper makes a solid contribution to multi-objective policy optimization and comparison-based learning.